# Pivotal role for skin transendothelial radio-resistant anti-inflammatory macrophages in tissue repair

Olga Barreiro[1]*[†], Danay Cibrian[1,2,3], Cristina Clemente[1], David Alvarez[4], Vanessa Moreno[1], Íñigo Valiente[5], Antonio Bernad[5], Dietmar Vestweber[6], Alicia G Arroyo[1], Pilar Martín[1], Ulrich H von Andrian[4,7], Francisco Sánchez Madrid[1,2,3]*

[1]Department of Vascular Biology and Inflammation, Centro Nacional de Investigaciones Cardiovasculares, Madrid, Spain; [2]Servicio de Inmunología, Hospital de la Princesa, Universidad Autónoma de Madrid, Madrid, Spain; [3]Instituto Investigación Sanitaria Princesa, Universidad Autónoma de Madrid, Madrid, Spain; [4]Department of Microbiology and Immunobiology, Harvard Medical School, Boston, United States; [5]Department of Cardiovascular Development and Repair, Centro Nacional de Investigaciones Cardiovasculares, Madrid, Spain; [6]Max Planck Institute of Molecular Biomedicine, Münster, Germany; [7]Ragon Institute of MGH, MIT and Harvard, Cambridge, United States

*For correspondence:
olga_barreiro@hms.harvard.edu
(OB); fsmadrid@salud.madrid.org
(FSM)

Present address: [†]Department of Microbiology and Immunobiology, Harvard Medical School, Boston, United States

Competing interests: The authors declare that no competing interests exist.

**Abstract** Heterogeneity and functional specialization among skin-resident macrophages are incompletely understood. In this study, we describe a novel subset of murine dermal perivascular macrophages that extend protrusions across the endothelial junctions in steady-state and capture blood-borne macromolecules. Unlike other skin-resident macrophages that are reconstituted by bone marrow-derived progenitors after a genotoxic insult, these cells are replenished by an extramedullary radio-resistant and UV-sensitive Bmi1[+] progenitor. Furthermore, they possess a distinctive anti-inflammatory transcriptional profile, which cannot be polarized under inflammatory conditions, and are involved in repair and remodeling functions for which other skin-resident macrophages appear dispensable. Based on all their properties, we define these macrophages as Skin Transendothelial Radio-resistant Anti-inflammatory Macrophages (STREAM) and postulate that their preservation is important for skin homeostasis.

## Introduction

Macrophages are myeloid cells highly specialized in pathogen clearance and antigen capture in lymphoid tissues, where they participate in the first line of immune defense against exogenous threats, extending a bridge between innate and adaptive immunity (*Varol et al., 2015*). Homeostasis is also maintained in a number of peripheral non-lymphoid tissues by the phagocytic, anti-microbial and tissue remodeling activities of specialized resident macrophages, including alveolar macrophages in lungs, interstitial histiocytes in connective tissue, osteoclasts in bone, microglia in brain, Kupffer cells in liver, peritoneal, intestinal and adipose tissue macrophages (*Murray and Wynn, 2011*). Likewise, macrophages contribute to the barrier function of the skin. Unlike migratory epidermal Langerhans cells and dermal dendritic cells, which are mobilized after topical antigen uptake to perform their tolerogenic or immunogenic role within the draining lymph nodes (*Henri et al., 2010*), skin-resident

**eLife digest** The skin forms an essential barrier that defends our body from external damage. For this reason, it is important to understand the complex mechanisms involved in healing wounds and maintaining healthy skin. This could allow us to find effective treatments for skin diseases such as atopic dermatitis and psoriasis.

Immune cells called macrophages can protect the body in different ways depending on the signals they receive. Their protective roles include killing microbes that may cause disease, and helping to repair damaged tissues. Barreiro et al. have now investigated the roles of the macrophages in the skin by means of a number of complementary techniques, including a method called intravital microscopy that can image cells in a living organism.

The experiments revealed that a division of labor exists among the macrophages that reside in the skin of mice. Some macrophages help to trigger inflammatory responses in the skin. These immune cells disappear after being exposed to ionizing radiation (such as that used to treat cancer) but can be replaced via a bone marrow transplant. Other macrophages that help to repair tissues can survive being exposed to ionizing radiation but cannot resist high levels of ultraviolet light. Both types of macrophages perform unique and essential roles, and both types are necessary for maintaining healthy skin.

Barreiro et al. also discovered that the skin macrophages that help to repair tissues have the ability to move into blood vessels and take up substances from the blood. A question for future investigation is whether the macrophages perform this scavenging process to trigger a protective immune response in the nearby skin.

macrophages remain in the dermis, contributing to pathogen clearance, tissue repair and the resolution of inflammation (*Pasparakis et al., 2014*).

A certain degree of heterogeneity among skin-resident macrophages has been previously reported relative to their ontogeny as well as to the functional specialization of specific subsets (*Abtin et al., 2014*; *Hoeffel et al., 2015*; *Schulz et al., 2012*; *Tamoutounour et al., 2013*). Herein, we define for the first time a clear dichotomy among the skin-resident macrophages based on their differential sensitivity to γ-irradiation. Radio-resistant and radio-sensitive skin macrophages are distinctively polarized already in steady state as revealed by transcriptional analysis. Consequently, these two macrophage subsets are functionally specialized in a cell-autonomous manner even though both subsets are exposed to mostly shared environmental cues. In particular, the radio-resistant macrophage subset comprises a novel type of perivascular macrophages that gain access to the vascular lumen, are highly phagocytic and possess anti-inflammatory properties even in the presence of pro-inflammatory stimuli. Moreover, they are renewed from a local Bmi1$^+$ progenitor and become outcompeted over time by bone marrow-derived resident macrophages. Finally, the preferential depletion of these skin transendothelial radio-resistant anti-inflammatory macrophages (STREAMs) using diphtheria toxin-OVA nanoparticles evidences their involvement in tissue repair and remodeling, as well as the inability of their radio-sensitive counterparts to act as functional surrogates.

## Results

### Dermal perivascular macrophages protrude into microvessels at steady-state

To visualize the microvascular network of the skin, we performed intravital imaging of the upper dermis of the mouse ear under steady-state conditions using a minimally invasive model based on confocal microscopy (*Auffray et al., 2007*). The vasculature was highlighted by injecting i.v. a non-permeable vascular tracer, high molecular weight (HMw; 2 MDa) TRITC-dextran. Remarkably, a population of perivascular cells became readily visible during the first hours after HMw dextran injection, suggesting that these mostly sessile cells were constantly taking up intravascular dextran (*Figure 1A* and *Video 1*). This observation was further confirmed by monitoring the uptake of two differently labeled HMw dextrans injected with a lapse of 24 hr (*Figure 1—figure supplement 1A*). Dextran

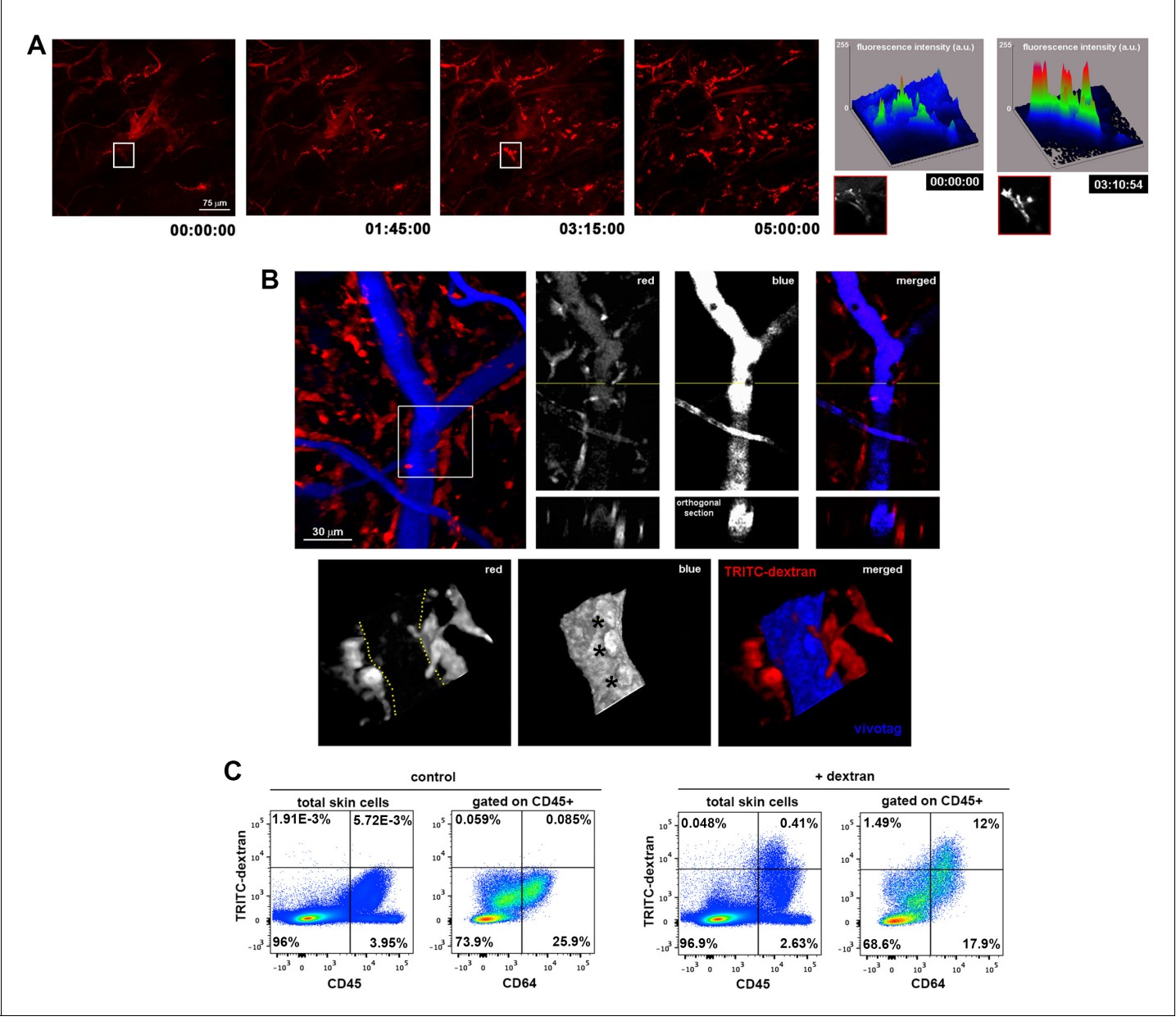

**Figure 1.** Dermal perivascular macrophages capture intraluminal dextran. (**A**) (Left) Representative frames from *Video 1* are shown. Briefly, image acquisition of ear mouse dermis started 30 min after i.v. injection of HMw TRITC-dextran and lasted for 5 hr. (Right) The surface plots show the fluorescence intensity of a representative cell at the starting time and 3 hr 15 min later (white insets in corresponding video frames; a.u., arbitrary units). (**B**) (Upper left) Three-dimensional (3D) rendering of the fluorescence signals obtained in a C57BL/6 ear dermis after i.v. administration of HMw TRITC-dextran (red, injected 16 hr before imaging) and vivotag (blue, injected at the time of imaging). (Upper right) The panels show a representative xy plane split into the different channels with an orthogonal section beneath, obtained along the yellow cross-section line. (Lower) The boxed region in the 3D rendering is shown tilted and at higher magnification. Yellow-dotted line marks endothelial perimeter in the red channel. Black asterisks mark void spaces corresponding to intravascular cells in the blue channel. (**C**) Phenotypic analysis of the dextran[+] cells in the ear skin using the pan-leukocyte marker CD45 and the macrophage marker CD64. Representative FACS dot plots of control and HMw TRITC-dextran-injected mice are shown (n = 4).

The following figure supplement is available for figure 1:

**Figure supplement 1.** Spatio-temporal analysis of endothelial-protruding macrophages under steady-state conditions.

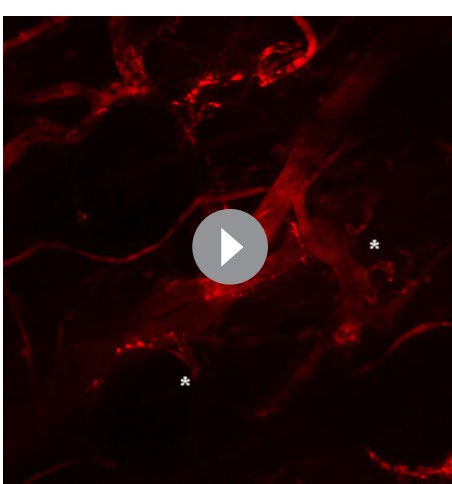

**Video 1.** Observation of intravascular dextran uptake by dermal non-migratory perivascular cells. Video sequence illustrating the progressive phagocytosis of intravascular dextran by perivascular cells (mostly sessile) under homeostatic conditions. A C57BL/6 animal was injected i.v. with HMw TRITC-dextran and immobilized by means of long-term anesthesia. Intravital imaging started 30 min after injection. The frames in the video sequence are maximal projections of a confocal z-stack (207 planes, 129.66 μm in depth) acquired every 5 min over 5 hr (only shown first 3 hr 15 min). White asterisks highlight cells of particular interest. The video also shows the clearance of dextran from the bloodstream and the absence of paravascular permeability under homeostatic conditions.

administration followed by the subsequent injection of a non-phagocytosable tracer (vivotag) revealed the presence of dextran$^+$ protrusions as non-stained regions in the vivotag channel projecting into the vessel lumen (*Figure 1B*). The protrusions were flapping inside the vessels, presumably deflected by the vascular flow, and even contacting circulating cells (*Figure 1—figure supplement 1B* and *Videos 2–4*). The phenotypic characterization of the TRITC-dextran$^+$ perivascular cells identified them mostly as macrophages (i.e., they were either CD45$^+$ F4/80$^{high}$ CD11c$^-$ CD11b$^+$ or CD45$^+$ F4/80$^{high}$ CD11c$^-$ CD64$^+$ cells by FACS [*Figure 1C* and *Figure 1—figure supplement 1C*] and CD68$^+$ cells by whole-mount immunofluorescence staining [*Figure 1—figure supplement 1D*]). Flow cytometry analysis of skin single-cell suspensions from 12-week-old animals showed that the 96.67% ± 1.02% of all TRITC-dextran$^+$ cells were CD45$^+$ and the 90.41% ± 2.06% of CD45$^+$ TRITC-dextran$^+$ cells corresponded to macrophages (CD64$^+$). In addition, TRITC-dextran$^+$ macrophages represented 41.28% ± 5.66% of total skin macrophages. Next, we performed a mesoscopic analysis of murine ears to characterize the spatial distribution of skin macrophages. This analysis revealed their preferential localization around dermal blood microvessels rather than lymphatics (*Figure 2—figure supplement 1A* and *Video 5*). Interestingly, TRITC-dextran$^+$ macrophages were mostly apposed to capillaries and venules, possibly because the dense smooth muscle layer wrapping arterioles prevented the access of macrophages to the arteriolar lumen (*data not shown*). Moreover, endothelial cells did not detectably capture dextran under our experimental conditions, ruling out the possibility of the endothelium making accessible the intravascular dextran to the perivascular macrophages by transcytosis (*Figure 2—figure supplement 1B*).

We next studied how the macrophage protrusions gain access to the vascular lumen. We detected dextran$^+$ macrophages embracing vessels from outside the pericyte sheath (visualized as GFP$^+$ cells around vessels using a nestin-eGFP reporter mouse), whereas their protrusions extended across the basement membrane (stained with anti-collagen IV) and between pericytes to reach the endothelial wall (whole-mount immunofluorescence stainings in *Figure 2A* and *Figure 2—figure supplement 1C*). We also visualized dextran$^+$ macrophage protrusions aligned with endothelial junctions in vivo by intravital microscopy by injecting i.v. an anti-CD31 antibody to stain the luminal surface of dermal microvasculature (*Figure 2—figure supplement 1D* and *Video 6*). Further analysis using orthogonal sectioning of whole-mount immunofluorescence stainings indicated that the macrophages insert their protrusions in the intravascular space through endothelial intercellular junctions (*Figure 2B*). Interestingly, TRITC-dextran uptake was minimal in VE-cadherin-α-catenin knock-in mice (*Figure 2C*). In these animals, endothelial adherens junctions are stabilized (*Schulte et al., 2011*), which prevents paravascular permeability and leukocyte paracellular but not transcellular diapedesis. The ability of dermal macrophages to access the intraluminal space is restricted in these animals, supporting the observation that macrophages can reach the vascular lumen crossing the endothelial junctions and excluding the possibility that their protrusions reach the vessel lumen via a transcellular route through endothelial cells. Finally, since the capture of dextran by skin myeloid cells is mediated by the mannose receptor (CD206) (*Wollenberg et al., 2002*), we could selectively stain the intraluminal protrusions of these dermal perivascular macrophages in vivo by injecting i.v. anti-CD206 and

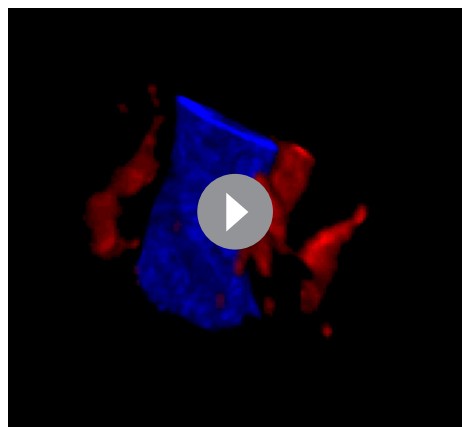

**Video 2.** Dermal perivascular cells protrude into dermal microvessels and contact intravascular leukocytes. Video animation of a 3D rendering from a z-stack acquired in vivo in the dermis of a mouse ear. The mouse was injected i.v. with HMw TRITC-dextran 16 hr before microscopic observation and injected i.v. with vivotag (as vascular tracer) just before the experiment. Intravital imaging with confocal sectioning (1 z-stack, 65 planes, 40.28 μm in depth) was performed. The image shows several perivascular dextran+ cells around a vessel highlighted in blue (vivotag staining). Remarkably, two adjacent perivascular cells introduce their protrusions into the vascular lumen to contact intravascular leukocytes, seen as holes not filled with the vascular tracer.

quantified the amount of endothelial-protruding macrophages by FACS (*Figure 2D*). Altogether these data indicate the existence of hitherto unidentified dermal perivascular macrophages able to protrude across endothelial junctions to reach the vascular lumen in homeostatic conditions.

## Dermal dextran+ macrophages are a radio-resistant subset of tissue-resident macrophages

In an attempt to visualize endothelial-protruding macrophages without dextran labeling, we first analyzed whether these macrophages selectively expressed CX3CR1, a marker associated to certain tissue-resident macrophage subsets including intestinal macrophages and microglia (*Varol et al., 2015*). However, dextran+ macrophages did not express fluorescent reporter protein in *Cx3cr1gfp/+* mice, neither in other myeloid reporter strains such as *Lyz2Cre:Rosa26YFP* and *Langerin-eGFP* (*Figure 3—figure supplement 1A*). Then, we generated chimeric animals by reconstituting lethally irradiated C57BL/6 mice with β-actin-GFP+ hematopoietic progenitors, since it has been previously reported the absence of radio-resistant macrophages in the skin (*Bogunovic et al., 2006*). Unexpectedly, dextran+ dermal macrophages did not express

GFP in these chimeras (*Figure 3A* and *Video 7*), indicating that they were not replaced by BM-derived macrophages after irradiation and suggesting that they could be a specialized radio-resistant subset of skin-resident macrophages never characterized before. In fact, we monitored the presence of perivascular macrophages able to phagocytose intravascular HMw dextran 7d after lethal γ-irradiation without BM reconstitution (*Figure 3—figure supplement 1B*). To further assess their radio-resistant nature, we reconstituted lethally irradiated CD45.2 host mice with CD45.1 BM progenitors. Reconstitution of the hematopoietic system in blood and lymphoid organs was complete after 12 weeks, with a negligible residual contribution of host cells to the myeloid compartment. In contrast, the chimerism rate in the skin myeloid subsets exhibited a higher contribution of the host populations, which could not be ascribed to blood reconstitution (*Figure 3—figure supplement 1C–D*). The host radio-resistant CD45.2+ macrophages efficiently captured HMw dextran under steady-state conditions and were found mostly positioned around microvessels (*Figure 3B* and *Figure 3—figure supplement 1E*). To validate this observation in a homeostatic context devoid of irradiation, the uptake of intravascular dextran was also analyzed in

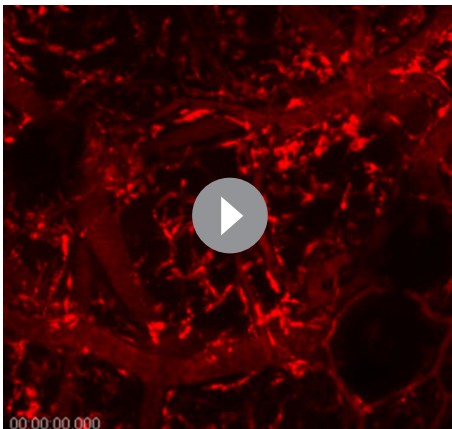

00:00:00.000

**Video 3.** Scanning movement of the protrusions of dermal perivascular cells. Frames represent maximal projections of a z-stack (9 sections acquired every 10 μm) obtained repeatedly (delay time 7 s 89 ms) over a short time period (9 min 30 s). The video shows the scarce motility of dextran-capturing cells.

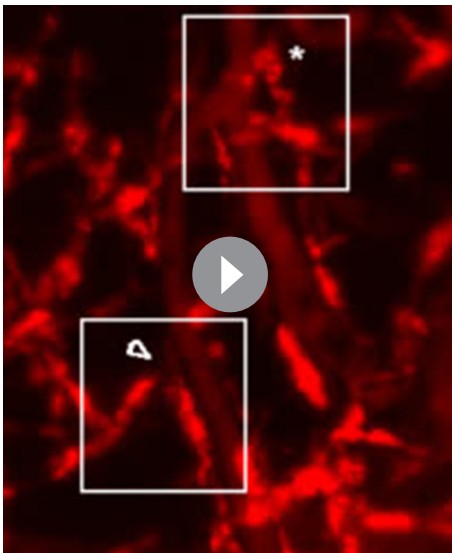

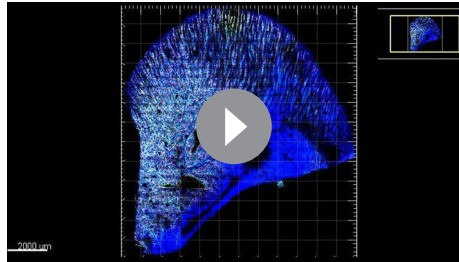

**Video 5.** Mesoscopic analysis of macrophage organization in relation to lymphatic and blood vessels. A whole mount-staining of a C57BL/6 ear dermis followed by tiled z-stack acquisition of the whole organ were performed. The video animation progresses from the mesoscopic analysis of the whole organ (ear) towards the detailed microscopic organization of lymphatics (LYVE-1$^+$, red), vascular endothelium (CD31$^+$, green) and tissue resident macrophages (CD68$^+$, white). Note the evident organization of macrophages around the complex endothelial network formed around hair follicles (minimal units of the skin) instead of around lymphatics. Blue corresponds to DAPI staining of nuclei, to facilitate observation of ear anatomy.

**Video 4.** Magnified detail from *Video 3*. Crop and zoom from *Video 3* for detailed observation. The white asterisk marks a cell with an intravascular protrusion that flaps inside the vessel due to blood flow and the arrowhead points to the rearward movement of a dextran-loaded phagocytic vesicle from the vessel and along the protrusion towards the perivascular cell body.

parabiotic mice maintained for 6 months with a shared circulation. The results obtained were similar to those of chimeric animals, inasmuch as only host macrophages captured intraluminal dextran (*Figure 3C*).

Interestingly, a recent study described the existence of a subset of GFP$^+$ perivascular macrophages in the DPE-GFP mouse strain, in which GFP is driven under the control of the distal and proximal CD4 enhancers and CD4 promoter (details on the generation of this mouse line described in [*Mempel et al., 2006*]). These GFP$^+$ macrophages provide guidance cues for the recruitment of inflammatory cells such as neutrophils during intradermic bacterial infection (*Abtin et al., 2014*). To analyze whether the dermal endothelial-protruding macrophages correspond to the GFP$^+$ macrophages described above, we injected HMw TRITC-dextran in DPE-GFP animals. Remarkably, ~80% of cells that acquired circulating HMw dextran were GFP$^-$ in the skin of DPE-GFP mice (*Figure 4A*). Furthermore, chimerism experiments in which we reconstituted lethally irradiated CD45.2 DPE-GFP hosts with CD45.1 hematopoietic progenitors revealed a dramatic decrease in the number of GFP$^+$ dermal macrophages one month after the genotoxic insult, indicating their radio-sensitive nature (*Figure 4B*). Altogether these results allow us to establish a clear dichotomy between DPE-GFP$^+$ and dextran-capturing perivascular macrophages.

## Radio-resistant macrophages derive from an extramedullary UV-sensitive source

In order to find the source for the skin-resident radio-resistant macrophages described in this work, we explored whether they are long-lived, endowed with self-renewal capacity, or derived from an extramedullary progenitor as alternative possibilities. We first tackled their sensitivity to UV irradiation, since other radio-resistant immune subset in the skin, the Langerhans cells (LC), is UV-sensitive (*Merad et al., 2002*). Chimeric CD45.1/CD45.2 (donor/host) animals were exposed to UVA-B irradiation (5 J/cm$^2$), and ear skin was analysed after one month. UV irradiation had a clearly negative effect on the survival and renewal of host compared with donor macrophages (*Figure 5A*), indicating that radio-resistant macrophages are UV-sensitive. However, the radio-resistant macrophages did not

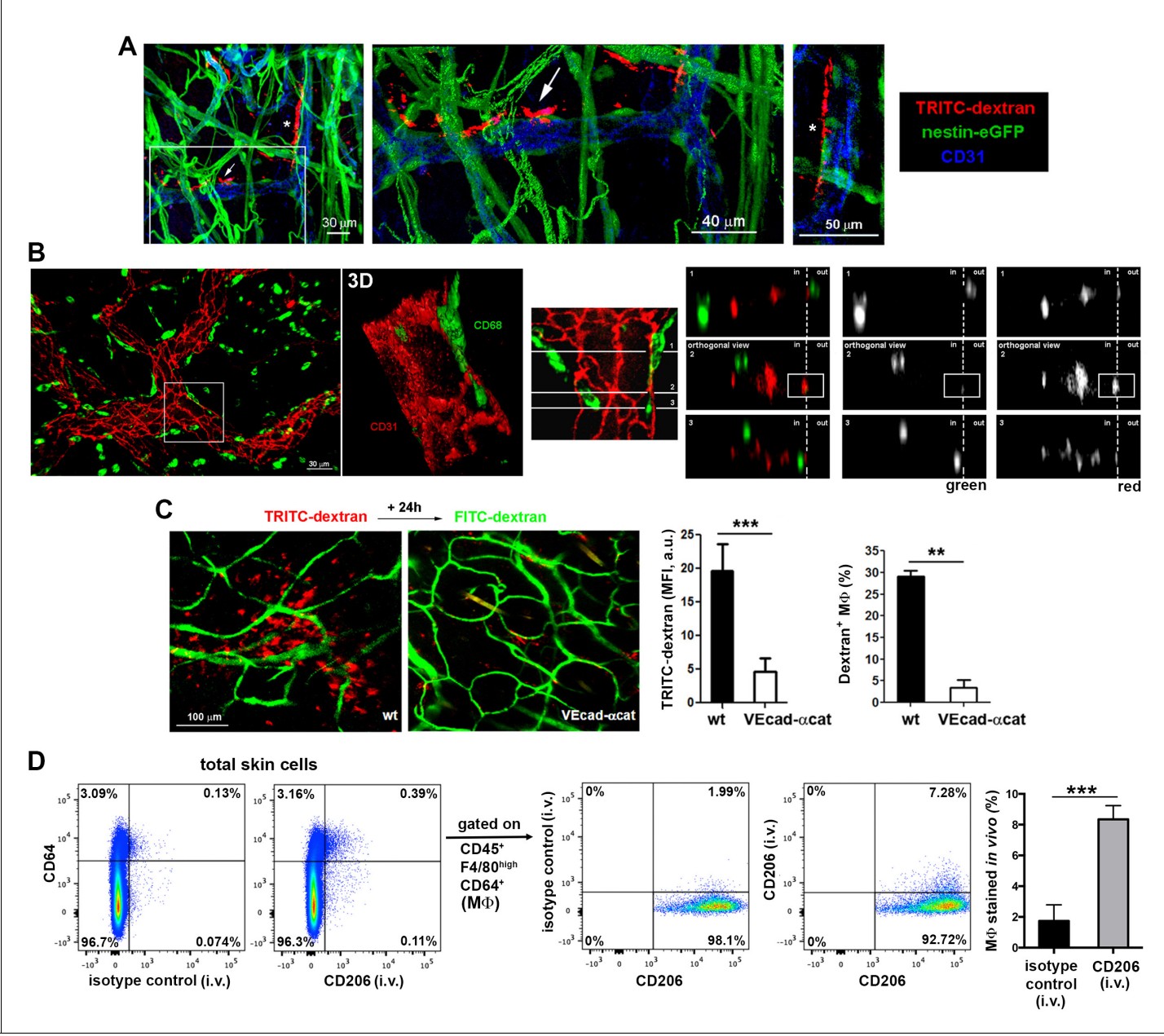

**Figure 2.** Dermal perivascular macrophages protrude across endothelial junctions into microvessels. (**A**) (Left) 3D reconstruction combining fluorescence signal and isosurface rendering of the ear dermis of a nestin-eGFP animal injected i.v. with HMw dextran. The inset is shown magnified in the central panel with a dextran$^+$ macrophage protruding into the vascular lumen highlighted with a white arrow. Another protruding macrophage, marked with an asterisk in the 3D reconstruction image, is shown in a zoomed single confocal plane on the right. (**B**) (Left) Whole-mount staining of a C57BL/6 ear using anti-CD31 (red) and anti-CD68 (green) antibodies. (Center) The inset is shown as a 3D rendering with higher magnification. (Right) The same inset was tilted and further analyzed by orthogonal sectioning. The white cross-section lines are localized along the cell body and the intravascular protrusion of the macrophage on the right side. (in, intraluminal; out, extravascular). (**C**) Comparison of dextran capture in wt vs. VE-cadherin-α-cat mice under steady-state conditions. Animals were injected i.v. with HMw TRITC-dextran and with HMw FITC-dextran 24 hr later, before imaging. (Left) Representative intravital images show the dermis of each phenotype, containing dextran$^+$ cells (red) and vasculature (green). (Center) The bar histogram represents the mean fluorescence intensity (MFI) of the TRITC-dextran signal ± SD obtained from five 20x fields of view of 2 animals of each phenotype (a.u., arbitrary units). (Right) Flow cytometry analysis of dextran capture by macrophages in wt and VE-cad-α-cat mice. Statistical significance was assessed by unpaired two-tailed Student's t-test (**p-value < 0.005, ***p-value < 0.001). (**D**) In vivo staining of the intraluminal protrusions of endothelium-protruding macrophages. Mice were injected i.v. with either an antibody against CD206 (mannose receptor, involved in dextran uptake) or an isotype control antibody and sacrificed 3 min later. Then, animals were perfused with PBS and ears were processed for FACS analysis. Single-cell suspensions were stained with anti-CD45, anti-F4/80, anti-CD64 and a different anti-CD206 clone. Representative FACS dot plots

*Figure 2 continued on next page*

*Figure 2 continued*

are depicted and the bar histogram shows the specific detection of macrophage staining in vivo (MΦ: macrophages). Data are mean values ± SD (n = 4). Statistical significance was assessed by unpaired two-tailed Student's t-test (\*\*\*p<0.001).

The following figure supplement is available for figure 2:

**Figure supplement 1.** Spatial organization of skin macrophages relative to lymphatic and blood vessels.

express langerin and are clearly distinguishable from epidermal and dermal langerin[+] dendritic cells (*Figure 3—figure supplement 1A*).

Next, we explored whether the radio-resistant macrophages proliferate in situ. We first treated chimeric animals intradermally with clodronate liposomes to deplete skin macrophages. Monitoring at 24 hr after injection showed depletion of both macrophage subsets, whereas after 7d we observed selective partial recovery of host macrophages, consistent with an in situ renewal of these cells (*Figure 5—figure supplement 1*). Then, we performed Ki-67 staining in the skin of host and donor macrophages from chimeric animals using LC as control cells endowed with self-renewal potential (*Chorro et al., 2009*) (*Figure 5B*). The results point out to host macrophages as the only cell type assayed unable to proliferate by itself in steady-state. We complemented the study analysing the incorporation of BrdU administered ad libitum during 8d in these subsets (*Figure 5C*). The results indicate that host radio-resistant macrophages have a slower turnover compared to BM-derived donor macrophages and LC. Altogether these findings suggest that skin radio-resistant macrophages are a long-lived differentiated population that originates from an in situ proliferating progenitor. Interestingly, the relative content of resident macrophages from donor and host changed dynamically over time in favor of the dominant BM-derived pool, which exhibits higher proliferative capacity (*Figure 5D*).

To identify the extramedullary source for dermal radio-resistant macrophages, we performed a fate mapping analysis in adult skin. First, we performed an analysis of the expression of a panel of stemness-related genes demonstrating that skin-resident macrophages possess no pluripotent potential per se in steady-state (*Figure 5—figure supplement 2A*). Then, we employed a reporter mouse (*Bmi1*-IRES-Cre-ERT2 Rosa26 YFP) driven by the promoter of *Bmi1*, gene involved in the maintenance of adult stemness (*Sangiorgi and Capecchi, 2008*), which enabled us to trace adult stem cells and their progeny in skin (*Figure 5—figure supplement 2B*). Macrophages contain very low levels of the *Bmi1* transcript (*Figure 5—figure supplement 2C*) and are devoid of *Bmi1* protein in steady-state (*Sienerth et al., 2011*). However, YFP[+] CD45[+] F4/80[high] CD11b[+] CD11c[-] macrophages were detected in the homeostatic skin of *Bmi1* reporter animals not earlier than 5d after i.p. injection of tamoxifen, suggesting that they are indeed derived from a Bmi1[+] progenitor (*Figure 5E*). Generation of YFP[+] macrophages was increased during macrophage replenishment after depletion with s.c. clodronate treatment (*Figure 5—figure supplement 2D*). Moreover, microscopy analysis of the dermis of these reporter animals detected perivascular YFP[+] cells protruding into vessels (*Figure 5F*). To exclude the contribution of BM-derived hematopoietic stem cells in this system, we generated chimeric CD45.1 C57BL/6 wt//*Bmi1*-IRES-Cre-ERT2

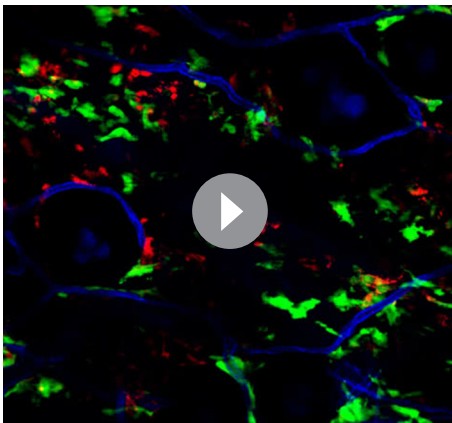

**Video 6.** In vivo imaging of dextran[+] macrophages aligned with endothelial junctions. The video sequence shows a confocal z-stack of the dermis of a mouse ear (65 planes, 40.28 μm in depth). The animal was chimeric (reconstitution with B6 ACTB-eGFP BM cells (green) in a C57BL/6 host) and was injected i.v. with HMw TRITC-dextran (red) 16 hr before imaging. For in vivo staining of endothelial junctions, an anti-CD31 antibody coupled to Alexa 647 (blue) was injected i.v. at the beginning of the experiment. The white circle and arrowheads highlight the area of interest, where the macrophage protrusions are posed in close contact with endothelial junctions.

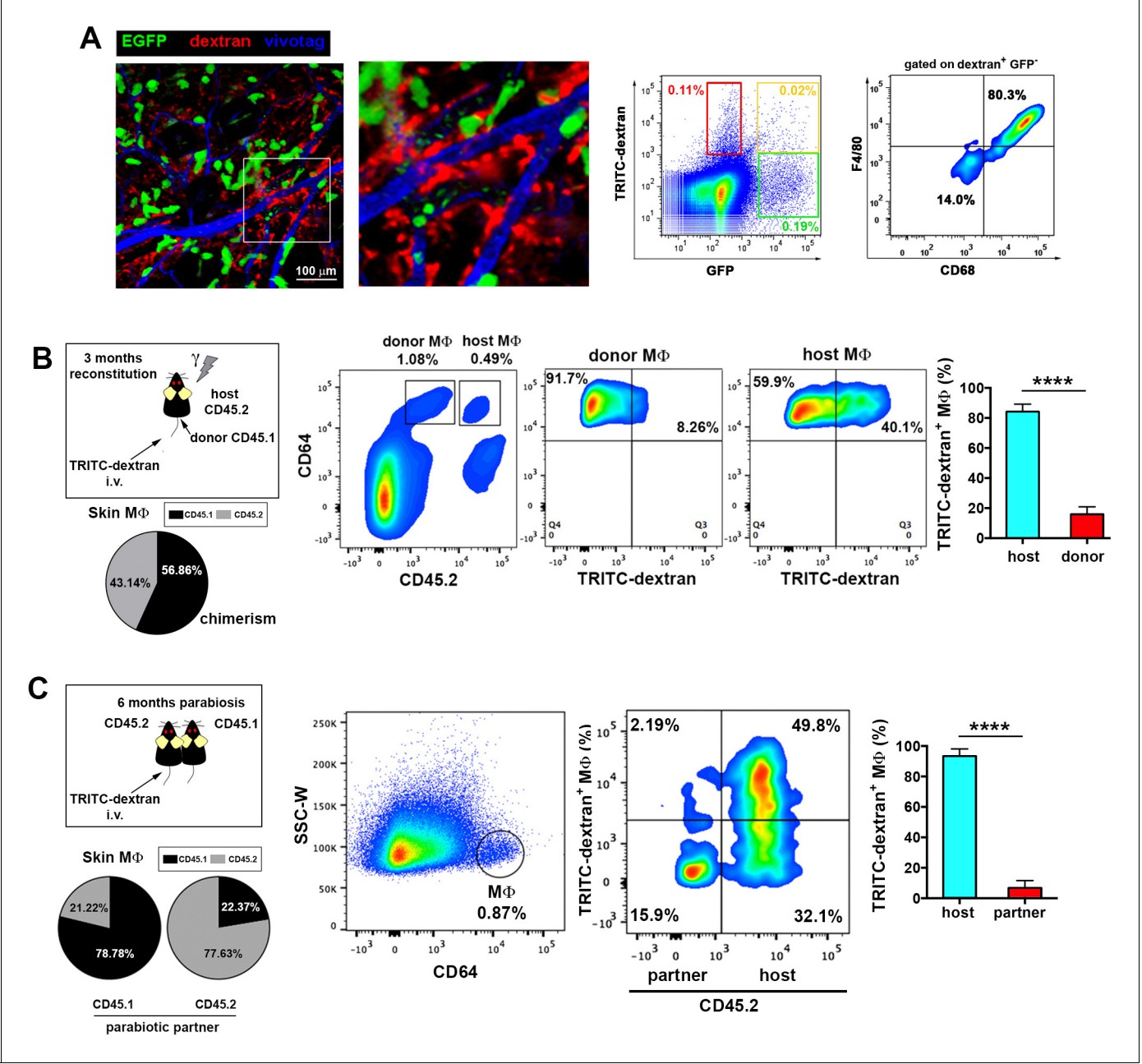

**Figure 3.** Endothelium-protruding macrophages are radio-resistant macrophages independent of BM supply. (A) (Left) Representative intravital image (maximal projection of a z-stack), with zoomed view aside, of the ear dermis of a chimeric B6 ACTB-eGFP (donor, green)/C57BL/6 (host) animal pre-treated with HMw TRITC-dextran (red) and injected with vivotag (blue) at the time of imaging. The 3D colocalization analysis between green and red signals was estimated (Pearson's coefficient in colocalized volume = 0.1074). (Right) Representative FACS dot plots showing the analysis of the TRITC-dextran+ (red), the GFP+ (green) and the double positive subsets in these chimeric animals as well as the macrophage content in the TRITC-dextran+ subset. (B) (Left) Averaged frequency of CD45.1+ (donor) and CD45.2+ (host) skin-resident macrophages in chimeric animals reconstituted for 3 months. (Right) Representative FACS analysis of the dextran+ subset in chimeric CD45.1 (donor)/CD45.2 (host) animals treated with HMw TRITC-dextran. The bar histogram shows the contribution of host and donor macrophages to the pool of dextran+ macrophages in the skin. Values are mean ± SD (n=4). Statistical significance was assessed by unpaired two-tailed Student's t-test (****p<0.0001). (C) Parabionts were injected i.v. with HMw TRITC-dextran and sacrificed 3d later. Ears were harvested and processed for FACS analysis. (Left) Averaged frequency of CD45.1+ and CD45.2+ skin-resident macrophages in each partner of the parabiotic pairs (n = 4) after 6 months of parabiosis. (Right) Representative dot plots and bar histogram showing the contribution of host macrophages and macrophages derived from partner to the pool of dextran+ macrophages in the skin. Values are mean ± SD (n = 4). Statistical significance was assessed by unpaired two-tailed Student's t-test (****p<0.0001).

*Figure 3 continued on next page*

*Figure 3 continued*

The following figure supplement is available for figure 3:

**Figure supplement 1.** Analysis of tissue-resident macrophage subsets in reporter and chimeric animals.

Rosa26 YFP CD45.2 (donor/host) mice. YFP$^+$ dextran$^+$ macrophages were detected in the skin of tamoxifen-treated HMw dextran-injected chimeric mice by FACS analysis (*Figure 5G–H*). Intravital experiments confirmed the existence of YFP$^+$ dextran$^+$ macrophages under steady-state conditions in chimeras lacking tamoxifen-inducible hematopoietic cells of BM origin (*Figure 5I*). Together these results confirm the in situ generation of radio-resistant macrophages from a skin pluripotent progenitor independent of BM.

## Phenotypic differences among skin-resident macrophage subsets

Our results strongly imply the coexistence of two distinct skin-resident macrophage pools in homeostasis based on their differential capacity to uptake circulating macromolecules, sensitivity to γ-irradiation and progenitor source. To further characterize these two populations, we performed a detailed phenotypic characterization of host radio-resistant and donor BM-derived macrophages in chimeric mice after 3 months of reconstitution. Both subsets were F4/80$^{high}$, CD11b$^+$ and expressed similar levels of other myeloid markers such as Ly6C and CD115, as well as of the prototypic macrophage markers CD68 and CD64-MertK (whose combined expression is associated with mature tissue macrophages [*Gautier et al., 2012*]) (*Figure 6A* and *Figure 6—figure supplement 1A*). Moreover, both subsets were CD11c$^-$ and Ly6G$^-$ (*Figure 6—figure supplement 1A*). There was also no consistent or sizeable difference in the expression of the angiogenesis-related receptor Tie-2, the costimulatory molecules CD40 and CD70 or MHC-I. However, we found a higher mean expression of the costimulatory molecule CD86 and the mannose receptor (CD206) in the host radio-resistant subset, whereas MHC-II was elevated in the donor BM-derived subset (*Figure 6A*). Nevertheless, the broad range of expression of these differential markers within each macrophage subset precluded their use to dissect accurately the heterogeneity found among the skin-resident macrophages (example provided for MHC-II in *Figure 6—figure supplement 1B*).

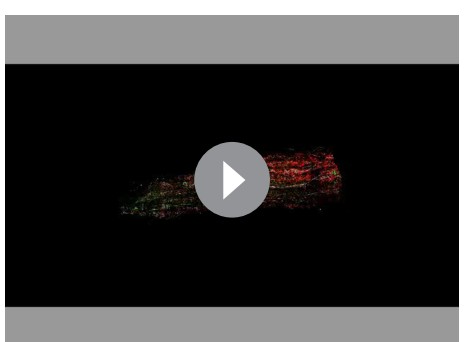

**Video 7.** Dextran$^+$ macrophages are not from BM origin. Video animation showing the maximal projection of a confocal z-stack obtained from the tail skin of a chimeric animal (B6 ACTB-eGFP(donor)/ C57BL6 (host)) injected i.v. with HMw TRITC-dextran. The animation alternatively shows the position of dextran$^+$ macrophages (red) and BM-derived hematopoietic cells (GFP$^+$, green) relative to vessels (CD31$^+$ in white, whole-mount staining) and finally highlights the lack of colocalization of red and green channels.

## Radio-resistant macrophages possess a distinctive anti-inflammatory gene expression profile

To find a specific marker for radio-resistant macrophages and gain further insight into the differences between the two tissue-resident macrophage subsets, we sorted donor and host macrophages from the skin of chimeric animals for high-resolution transcriptome profiling, using deep-sequencing technology (RNA sequencing (RNA-Seq)) (*Wang et al., 2009*). Of 17,741 genes compiled, we found significant differences in 744, 135 corresponding to protein coding sequences with similar normalized expression levels across replicates (*Figure 6B–C*, and *Supplementary file 1*). A comprehensive Gene Set Enrichment Analysis (GSEA) of all genes expressed in at least one cell type showed 36 Kegg pathways upregulated in host radio-resistant macrophages (14 of them enriched at a nominal p-value < 1%) vs. 149 in donor macrophages (26 pathways enriched at a nominal p-value < 1%). The GSEA analysis confirmed striking differences in metabolism between the

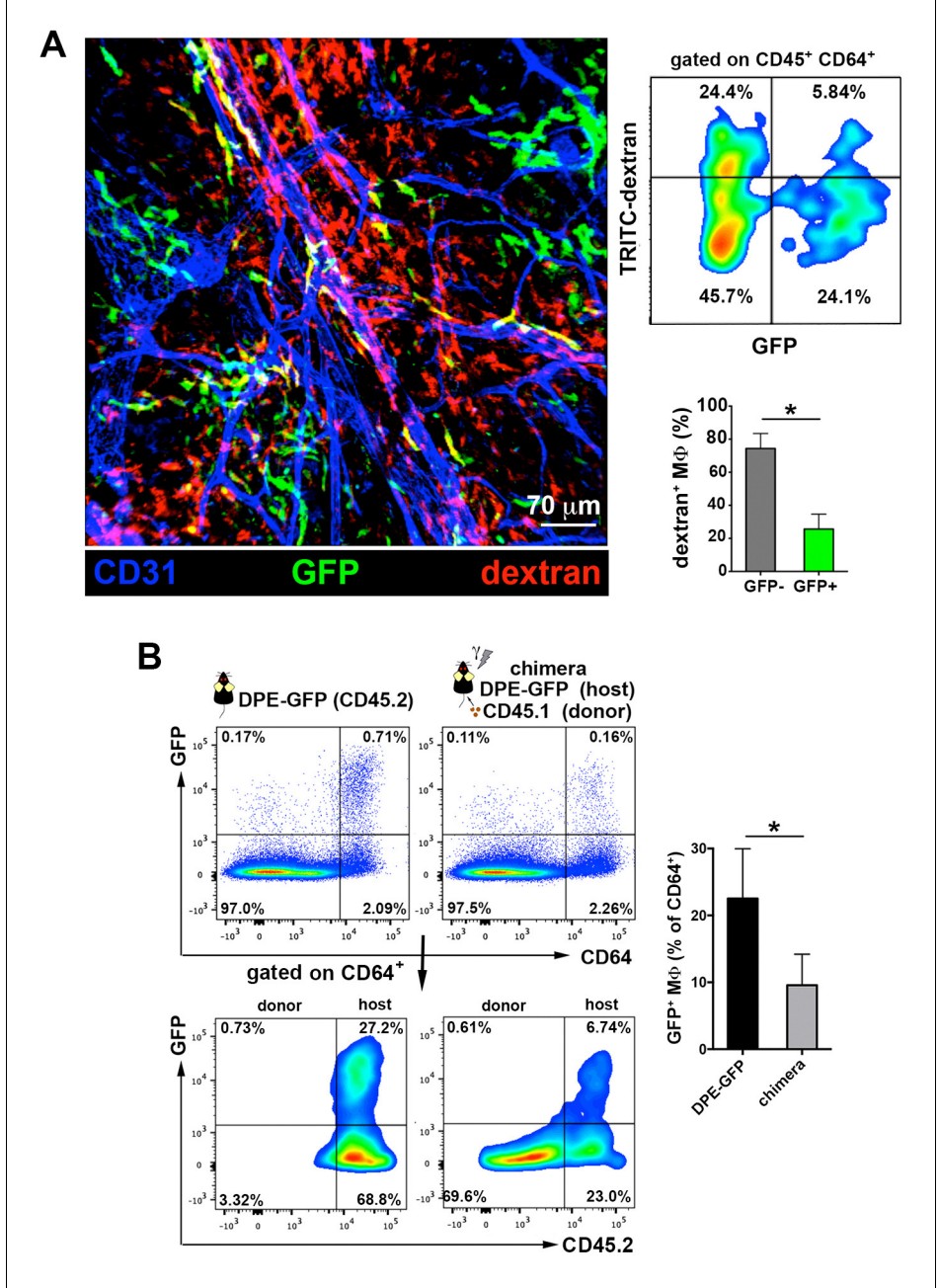

**Figure 4.** The perivascular endothelium-protruding macrophages are distinct from the perivascular DPE-GFP+ macrophages. (**A**) (Left) Maximal projection of the ear dermis of a DPE-GFP animal injected i.v. with HMw TRITC-dextran, fixed and stained with anti-CD31. The 3D colocalization analysis of the green (GFP+ perivascular MΦ) and red (dextran+ perivascular MΦ) signals was estimated (Pearson's coefficient in colocalized volume = 0.1901). (Upper right) FACS analysis of macrophages in ear skin from DPE-GFP animals injected i.v. with HMw TRITC-dextran. (Lower right) Bar histogram showing contribution of GFP- and + subsets to the pool of dextran+ macrophages. Values are mean ± SD (n = 3). Statistical significance was assessed by unpaired two-tailed Student's t-test (* p<0.05). (**B**) DPE-GFP CD45.2 animals were lethally irradiated and reconstituted with wt congenic CD45.1 BM. Animals were sacrificed 1 month later and the content of skin GFP+ MΦ in their skin was analyzed by FACS and compared with untreated DPE-GFP CD45.2 animals. The bar histogram on the right shows the % of GFP+ macrophages out of total skin macrophages. Data are mean ± SD (n = 4). Statistical significance was assessed by unpaired two-tailed Student's t-test (* p<0.05).

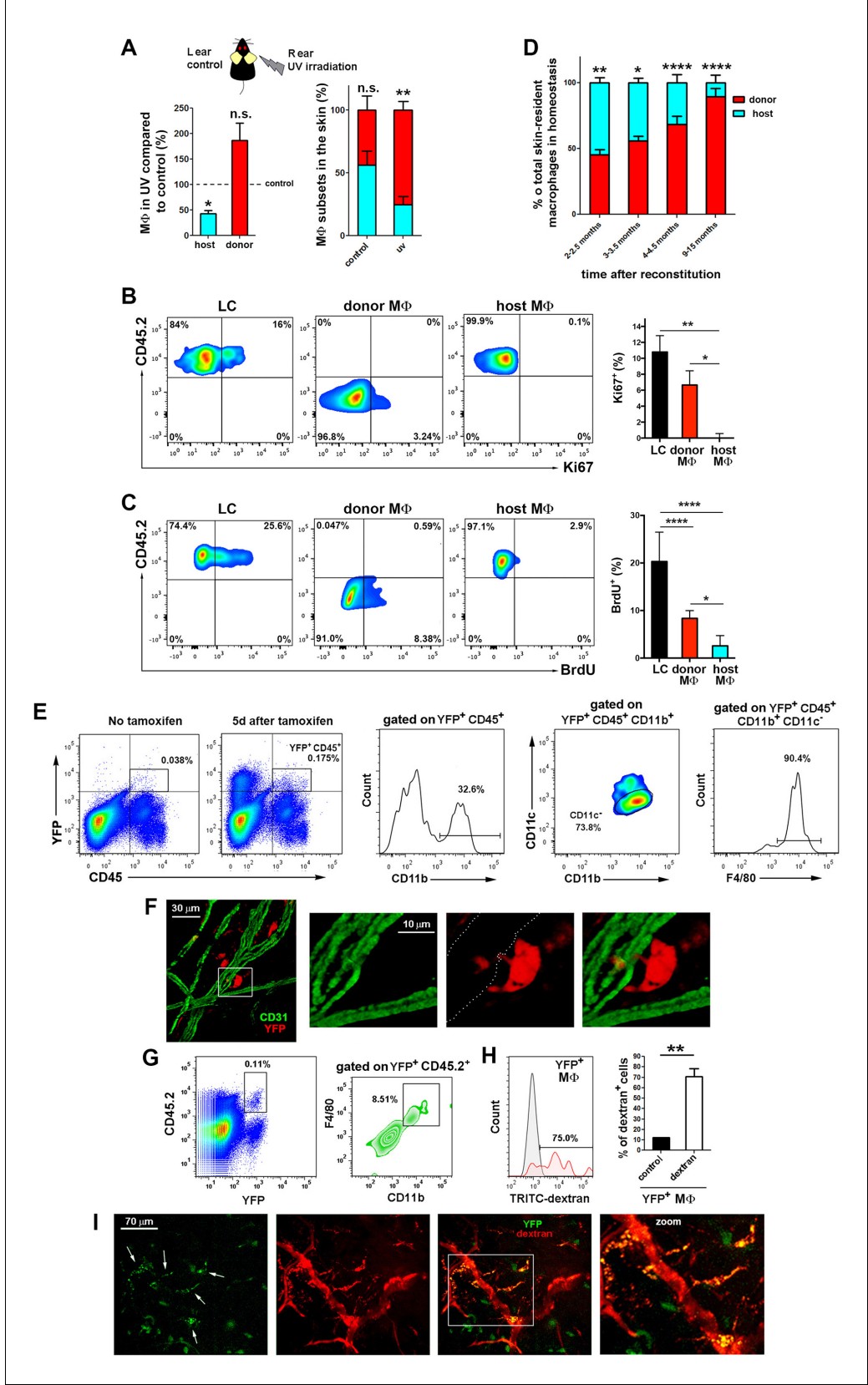

**Figure 5.** The skin radioresistant macrophages are an UV-sensitive population endowed with in situ renewal potential from an extramedullary progenitor. (**A**) Ears from chimeric animals (CD45.1(donor)/CD45.2(host)) were

*Figure 5 continued on next page*

*Figure 5 continued*

analyzed by flow cytometry 4 weeks after irradiation with UV-A/B or no irradiation. The left bar histogram shows the percentage of remaining macrophages of each haplotype after UV irradiation relative to untreated controls (n = 3/group). The second bar histogram depicts the relative content of each subset out of the total pool of tissue-resident macrophages in treated and non-treated animals. Data are means ± SEM. Statistical significance was assessed by two-way ANOVA analysis with Sidak's post-test (left) and one-sample t-test referred to value = 100 (right) (n.s. not significant, **p-value < 0.01). (B) Ki-67 staining reveals the lack of self-renewal capacity of radioresistant host macrophages in steady-state. Chimeric mice (CD45.1-donor/CD45.2-host) reconstituted for 3 months were sacrificed and skin processed and stained with anti-Ki-67 for FACS analysis. Tissue-resident macrophages from host (CD45.2$^+$ and CD64$^+$) and donor (CD45.1$^+$ and CD64$^+$) as well as Langerhans cells (LC, CD45.2$^+$ CD326$^+$ and MHC-II$^+$, used as control subset endowed with self-renewal capacity) were analyzed. Values are mean ± SEM (n = 4). Statistical significance was assessed by one-way ANOVA with Dunnett's post-test (*p-value < 0.05, **p-value < 0.005). (C) BrdU tracing revealed the slower turnover rate of host radioresistant macrophages. Chimeric mice (CD45.1-donor/CD45.2-host) were treated with BrdU in drinking water ad libitum. Animals were sacrificed 8 d later and skin processed for BrdU staining. Tissue-resident macrophages from host and donor as well as Langerhans cells were analyzed. Values are mean ± SD (n = 4). Statistical significance was assessed by one-way ANOVA with Tukey's post-test (**p-value < 0.005, ****p-value < 0.0001). (D) Fluctuations over time in the relative content of host and donor macrophages in the homeostatic skin of chimeric animals (n = 5/group). Data are mean ± SD. Statistical significance was assessed by two-way ANOVA with Sidak's post-test (*p-value < 0.05, **p-value < 0.01, ***p-value < 0.005, ****p-value < 0.0001). (E) Ears from tamoxifen-treated *Bmi1*-IRES-Cre-ERT2 Rosa26 YFP reporter mice were analyzed by flow cytometry 5d after treatment. Representative FACS analysis of the identified YFP$^+$CD45$^+$ fraction is shown. (F) Whole-mount staining of an ear from a tamoxifen-treated *Bmi1*-IRES-Cre-ERT2 Rosa26 YFP mouse. The boxed area highlights a YFP$^+$ perivascular cell protruding into a vessel, shown at high magnification in the accompanying panels. The white-dotted line marks the vasculature in the red channel. (G) *Bmi1*-IRES-Cre-ERT2 Rosa26 YFP CD45.2 host mice were chimerized with bone marrow from CD45.1 wt donors. Ears of these chimeric mice were analyzed by FACS and a subset of YFP$^+$CD11b$^+$F4/80$^{high}$ myeloid cells was detected. (H) FACS analysis of *Bmi1*-IRES-Cre-ERT2 Rosa26 YFP chimeric mice injected with HMw dextran. The bar histogram represents the percentage of YFP$^+$ dextran$^+$ macrophages respect to untreated control (n = 4). Data are mean ± SD. Statistical significance was assessed by unpaired two-tailed Student's t-test (**p-value < 0.01). (I) Representative intravital images of the ear of a chimeric mouse generated as in G, and injected with HMw TRITC-dextran. White arrows in the green channel mark YFP$^+$dextran$^+$ perivascular cells.

The following figure supplements are available for figure 5:

**Figure supplement 1.** Selective recovery of host macrophages in chimeric animals treated with clodronate liposomes intradermally.

**Figure supplement 2.** Absence of stemness gene expression in skin resident macrophages in steady-state.

two resident macrophage populations in homeostatic skin (*Figure 6—figure supplement 2A* and *Supplementary file 2*). Host radio-resistant macrophages displayed a metabolic gene profile based on fatty acid oxidation and oxidative phosphorylation, whereas donor BM-derived macrophages showed a glycolytic gene profile. Such metabolic differences found at the gene expression level were confirmed by assessing mitochondrial activity ex vivo. Direct staining of the skin with an indicator of mitochondrial membrane potential (MitoTracker Orange CMTMRos) revealed elevated mitochondrial function in host radio-resistant macrophages, although both macrophage subsets contained comparable amounts of mitochondria per cell (as indicated by similar levels of Tomm20, a translocase of the outer mitochondrial membrane) (*Figure 6—figure supplement 2B*). Thus, the metabolic profiles of donor and host macrophages in homeostatic skin mirrored the metabolic reprogramming of pro-inflammatory classically activated and anti-inflammatory alternatively activated macrophages (*Galvan-Pena and O'Neill, 2014*), respectively.

In addition to metabolic differences, we further validated the differential expression observed for several immune-related genes using a microfluidic-based multiplex qRT-PCR. Donor-derived macrophages showed increased expression of genes such as *H2-Eb1* (coding for MHC-II), *Ccr2*, *Ilβ*, *Il6* and *Il10*. Importantly, the latest three genes code for cytokines related to the pro-inflammatory response in macrophages (*Rodriguez-Prados et al., 2010*). Host macrophages showed upregulated

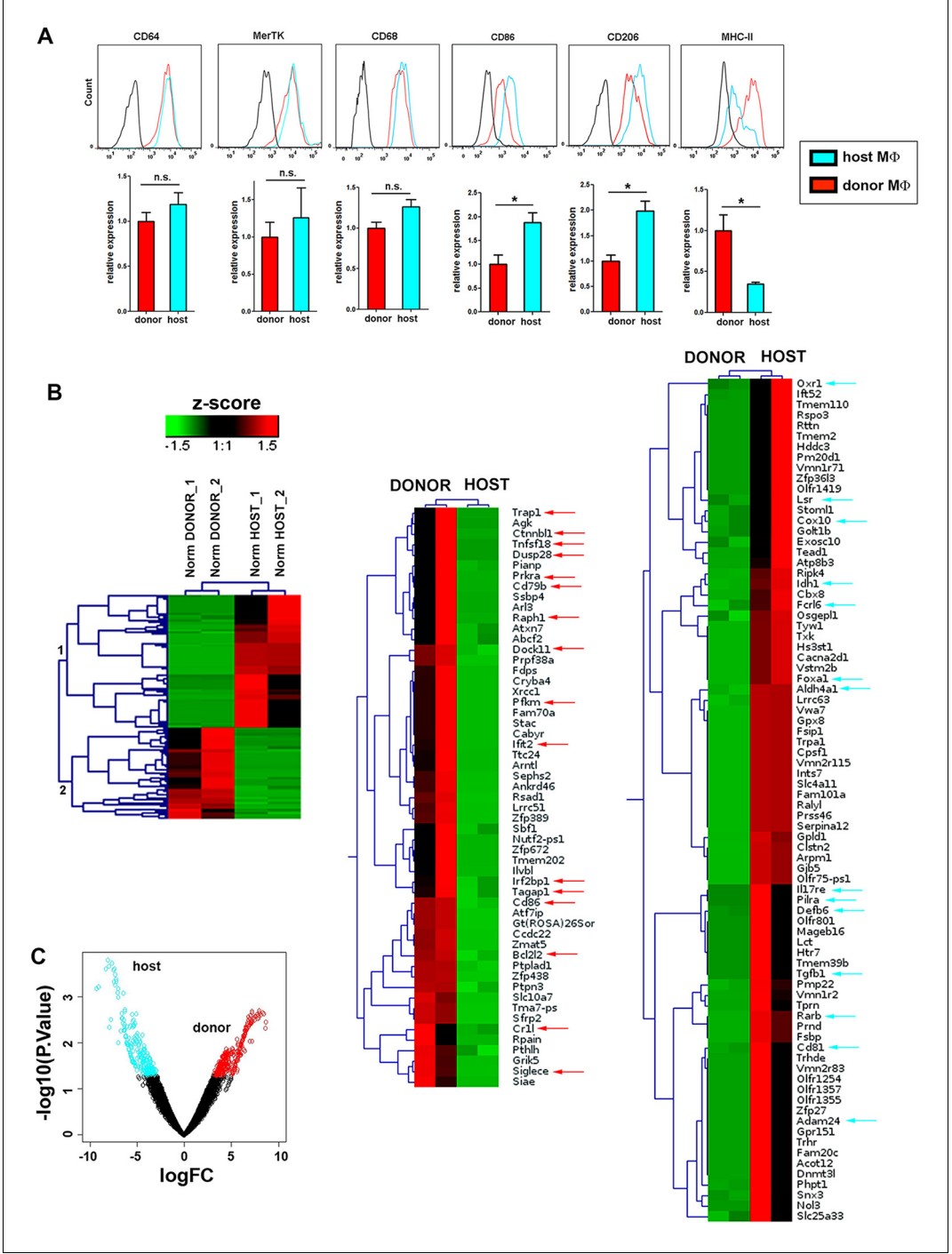

**Figure 6.** Phenotypic and RNASeq analyses of donor and host skin-resident macrophages from chimeric animals. (**A**) Flow cytometry analysis of the surface expression of macrophage-specific markers (CD64, MerTK, CD68), as well as CD86, CD206 and MHC-II in donor and host macrophages from chimeric animals in steady-state. Values correspond to the normalized average GeoMean ± SD (n = 3). Statistical significance was assessed by unpaired two-tailed Student's t-test (n.s. not significant, *p-value < 0.05, **p-value < 0.01). (**B**) (Left) Compact view of the hierarchical clustering of the normalized expression profiles of the protein coding sequences with p-value <= 0.05 and consistent expression levels across replicates. (Center and right) Detailed view of the hierarchical clustering showing gene annotation. Arrows highlight relevant genes upregulated in host (blue) and donor (red). (**C**) Vulcano plot representing the log2FC vs –log10 p-value transformation. Highlighted points correspond to genes with a p-value smaller than 0.05 (blue for genes more expressed in host than donor and red for genes with the opposite expression profile).

*Figure 6 continued on next page*

*Figure 6 continued*

The following figure supplements are available for figure 6:

**Figure supplement 1.** Extended phenotypic analysis of host and donor skin-resident macrophages.

**Figure supplement 2.** GSEA of the RNASeq data.

expression of *Cd86, Il15 and Hmox* among others (*Figure 7A*). Accordingly, we detected elevated protein expression of heme oxigenase-1 (encoded by *Hmox*) and of other anti-inflammatory markers such as arginase I (*Figure 7B*) as well as higher production of TGF-β (*Figure 7C*) in host macrophages in resting conditions.

Importantly, the immune gene signatures of the dermal macrophages expressing low MHC-II and never exposed to γ-irradiation and of the remaining dermal macrophages in the skin 7d after lethal γ-irradiation showed lack of upregulation of proinflammatory genes (*Figure 7—figure supplement 1A*), correlating with the observations made in radio-resistant host macrophages in chimeric animals. In contrast, dermal macrophages expressing high MHC-II and never exposed to γ-irradiation possess a pro-inflammatory profile as observed for donor BM-derived macrophages in chimeric animals (*Figure 7—figure supplement 1A*). These results suggest that the prior lethal irradiation to generate BM chimeras did not alter the immune gene expression profiles studied herein.

Hence, our results indicate that the radio-resistant subset of skin-resident macrophages is skewed towards an anti-inflammatory phenotype already in steady-state and predict a certain degree of functional specialization for them. We have coined this novel subset of macrophages as Skin Trans-endothelial Radio-resistant Anti-inflammatory Macrophages (STREAM) based on all the specific hallmarks found in our study.

## Skin anti-inflammatory macrophages are refractory to inflammatory stimuli

The divergent immune-related transcriptional profiles observed in the two distinct macrophage subsets coexisting in homeostatic skin prompted us to investigate whether they are committed to specific pro- or anti-inflammatory functions in a cell-autonomous manner and independently of polarizing stimuli. To examine this possibility, we isolated both macrophage subsets treated them ex vivo with LPS, a stimulus for which both subsets expressed the prerequisite receptors (*Figure 7—figure supplement 1B*). Interestingly, the pro-inflammatory macrophages became activated and produced substantial levels of CCL4, CCL5, CXCL1, TNF-α and IL-6 (among other pro-inflammatory mediators examined) in comparison to control conditions, whereas the anti-inflammatory subset did not produce significant amounts of these cytokines and chemokines in response to LPS (*Figure 7D*). However, the anti-inflammatory macrophages produced IL-10 in response to IL-4 stimulation (*Figure 7E*). These results suggest that the anti-inflammatory macrophages have a dearth of functional plasticity, being refractory to pro-inflammatory challenges but responsive to anti-inflammatory stimuli. They also emphasize the fact that the different subsets of skin-resident macrophages could be already committed even since steady-state to perform non-redundant functions.

## Selective depletion of STREAMs impairs skin repair

To explore the function of STREAMs, we searched for a method to deplete them selectively from the skin. For this purpose, we studied the capacity of STREAMs to capture nanoparticles coated with ovalbumin (OVA), another ligand for the mannose receptor (*Burgdorf et al., 2006*) that is highly expressed on their surface. In this manner, we ensured the active capture of the particulate OVA (OVA-coated fluorescent nanoparticles (FITC fluospheres), Φ=0.2 μm) from the bloodstream, preventing its passive diffusion into the perivascular tissue that could allow for extravascular phagocytosis. We first observed that OVA fluospheres injected i.v. were specifically captured by skin macrophages (*Figure 8A*), and this process was abrogated after the depletion of skin macrophages by s.c. clodronate treatment (*Figure 8B*). Intravital imaging followed by flow cytometry evaluation confirmed that the macrophages able to capture OVA fluospheres were dextran[+] in C57BL/6 wt animals (*Figure 8C*, *Figure 8—figure supplement 1A* and *Videos 8–9*) as well as radio-resistant in

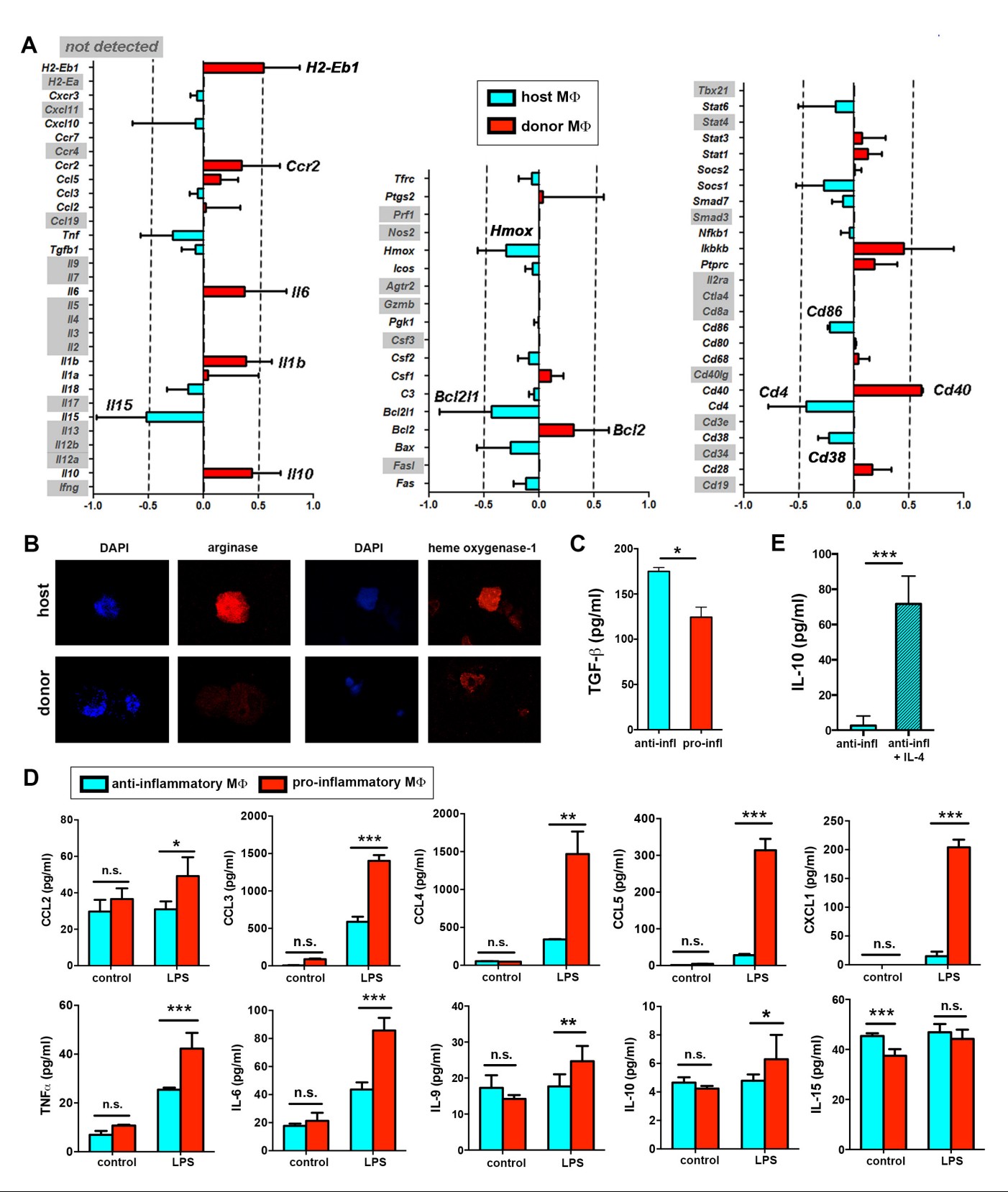

**Figure 7.** Immune transcriptional profile and polarization potential of STREAMs. (**A**) Comparative qPCR analysis of immune response-related genes in donor and host skin macrophages from chimeric mice. Data represent average logFC values ± SEM (n = 4). (**B**) Immunostaining of anti-inflammatory

*Figure 7 continued on next page*

*Figure 7 continued*

markers arginase I and heme oxygenase-1 in cytospin samples of isolated macrophage subsets (donor and host from chimeric animals). (**C**) ELISA quantification of TGF-β secretion by anti-inflammatory and pro-inflammatory macrophages at steady-state. Data are mean value ± SD (n = 4). Statistical significance was assessed by unpaired two-tailed Student's t-test (*p<0.05). (**D**) Sorted pro-inflammatory and anti-inflammatory MΦ were left untreated or treated with 1 ng/ml LPS for 24 hr. Then, a multiplex flow cytometry-based analysis of culture supernatants was performed. Data are mean ± SD (n = 3–5). Statistical significance was assessed by two-way ANOVA with Bonferroni's post-test (n.s. not significant, *p-value < 0.05, ** p-value < 0.01, ***p-value < 0.005). (**E**) ELISA quantification of IL-10 secretion by anti-inflammatory macrophages (50.000 cells/100 μl RPMI medium) at steady-state and after IL-4 stimulation (20 ng/ml) for 24 hr. Data are mean value ± SD (n = 4). Statistical significance was assessed by unpaired two-tailed Student's t-test (***p-value < 0.005).

The following figure supplement is available for figure 7:

**Figure supplement 1.** Gene profile of skin macrophages from non-treated and γ-irradiated animals.

CD45.1/CD45.2 (donor/host) C57BL/6 chimeras (*Figure 8D–E*). The availability of blood-borne nanoparticles in peripheral circulation is limited by the filtering action of mainly the liver and spleen, and therefore the amount of OVA-coated beads captured by skin STREAMs was low compared with that of splenic macrophages (*Figure 8—figure supplement 1B*); however, this amount was increased in γ-irradiated and splenectomized animals (*Figure 8—figure supplement 1A*).

Based on the specific ability of STREAMs to capture OVA nanoparticles, we next set up a method to deplete them selectively by administering i.v. OVA-coated nanoparticles (NP) adsorbed with diphtheria toxin (DT). In contrast to the i.v. administration of clodronate liposomes, which failed to eradicate skin STREAMs (*data not shown*), this tailored experimental strategy depleted preferentially STREAMs preserving most of the other skin macrophages that had no access to the blood-borne toxin-coated NPs (*Figure 9—figure supplement 1*). Then, the potential anti-inflammatory function of STREAMs in tissue repair was interrogated using a model of wound healing in combination with DT-OVA-NP treatment. Animals were treated either with OVA-NP or DT-OVA-NP (2.5 μl/g per dose) 1d before and 2d after wounding and were sacrificed at d5, observing a clear decrease in the macrophage population (*Figure 9A,D*). At this mid-stage of repair, macrophage depletion was associated with a significant delay in the healing process (measured as % of area of initial wound) (*Figure 9B*) and the histological analyses revealed a marked reduction in the formation of granulation tissue, concomitant reduced angiogenesis as well as delayed re-epithelialization (*Figure 9C–D*). Next, we administered a sole dose (2.5 μl/g) of DT-OVA-NP 2d after wounding and monitored wound closure till d9. In this manner, the macrophage depletion was attained after the initial inflammatory phase of wound healing (*Gurtner et al., 2008*). The Masson trichrome staining demonstrated a clear defect in collagen deposition in the animals treated with DT-OVA-NP (*Figure 10A*). This defect correlated with the disordered distribution of myofibroblasts in these animals observed with the α-smooth muscle actin (SMA) staining (*Figure 10B*). These results contrasted with the abundant collagen deposition and the parallel organization of myofibroblasts in OVA-NP-treated animals (*Figure 10A–B*). This noticeable phenotype observed in macrophage function-related parameters indicates that STREAMs could exert a prominent role as the macrophages involved in orchestrating tissue remodeling and repair in skin, as anticipated by their anti-inflammatory gene and protein expression profile.

Next, we performed wound healing experiments abrogating monocyte infiltration in the skin and depleting the radiosensitive subset of skin macrophages to further study the role of STREAMs in such scenario. We first analyzed the healing process in *Ccr2$^{-/-}$* animals, wherein the repopulation of skin from blood monocytes is compromised (*Boring et al., 1997*; *Serbina and Pamer, 2006*). Interestingly, STREAMs are CCR2$^-$ (*Figure 10—figure supplement 1A*) and are normally localized around dermal vessels in *Ccr2$^{-/-}$* mice (*Figure 10—figure supplement 1B*). Importantly, we did not observe substantial differences in wound closure, collagen deposition, and myofibroblast distribution in *Ccr2$^{-/-}$* mice compared to *Ccr2$^{+/+}$* animals (*Figure 10—figure supplement 1C–D*). Next, we benefited from the radio-resistant nature of STREAMs to deplete skin radio-sensitive macrophages by lethal γ-irradiation and prevented their replenishment by limiting the affluence of peripheral blood monocytes after reconstitution with *Ccr2$^{-/-}$* hematopoietic progenitors (*Willenborg et al., 2012*) (*Figure 10—figure supplement 1E*). We could not find differences in the healing capacity of

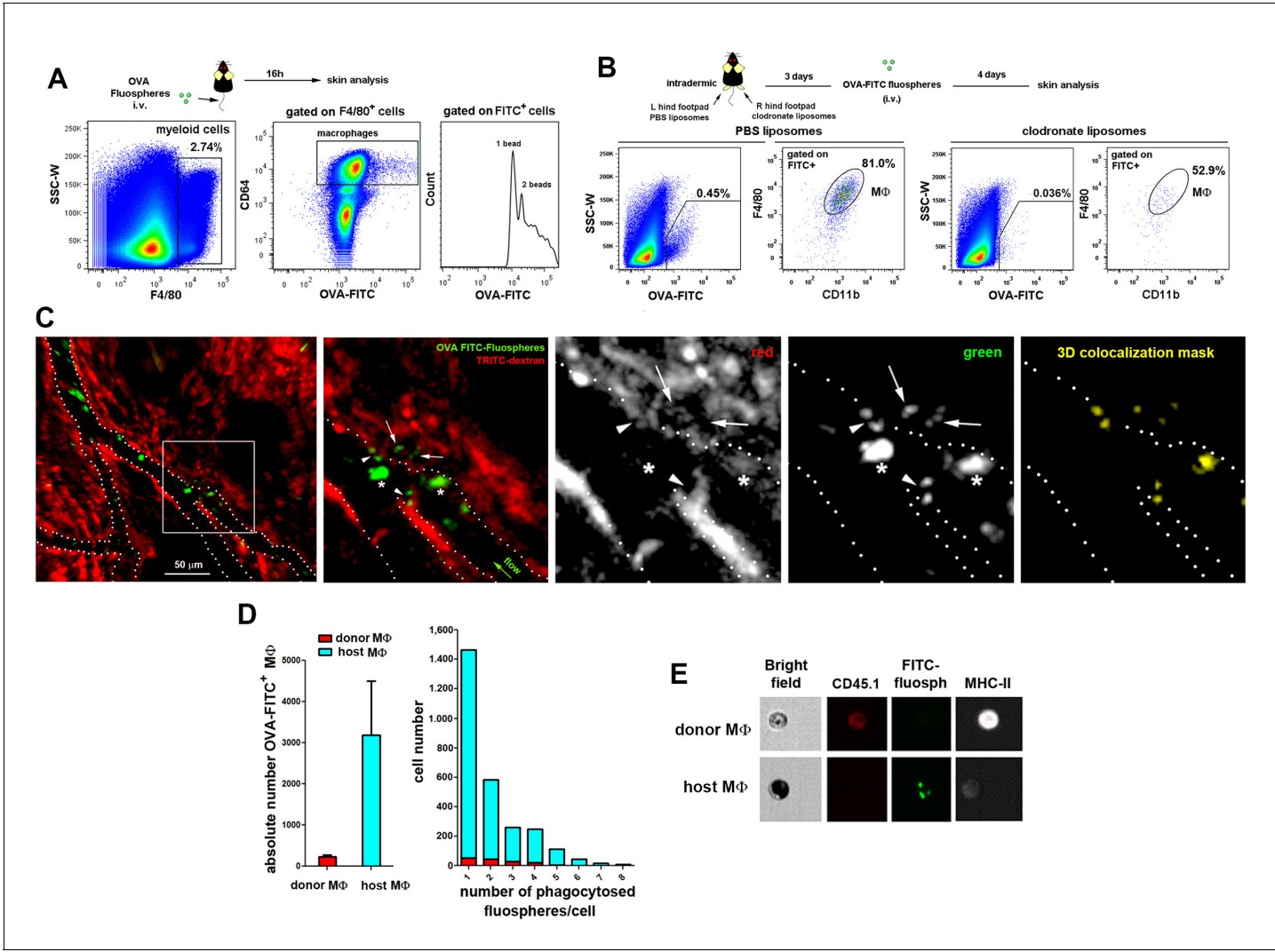

**Figure 8.** STREAMs capture blood-borne OVA nanoparticles at steady-state. (**A**) Representative flow cytometry analysis of OVA-FITC fluosphere capture by skin macrophages in C57BL/6 mice. The peaks observed in the histogram correspond to the number of fluospheres uptake per cell. (**B**) Mice were intradermally injected with PBS (control) or clodronate liposomes in the hind footpads and were injected i.v. with OVA-adsorbed FITC fluospheres 72 hr later, followed by analysis after 4d. Representative dot plots show the amount of FITC+ cells for each treatment and histograms show the proportion of MΦ in the FITC+ fraction. (**C**) (Left) A C57BL/6 mouse was first injected i.v. with HMw TRITC-dextran and with OVA-FITC fluospheres 16 hr later. Imaging began 24 hr after the last injection. A representative frame of the experiment extracted from *Video 8* with zoomed detail aside is shown. White dotted lines depict vessel walls. White asterisks mark FITC+ intravascular monocytes that have phagocytosed OVA fluospheres. White arrowheads point to intraluminal particles retained by dextran+ STREAMs and white arrows point to extravascular particles already phagocytosed by STREAMs. (Center and right) Split channels are shown with higher magnification as well as a 3D colocalization mask of both channels. (**D**) (Left) Absolute number of macrophages of each haplotype (CD45.1 (donor) or CD45.2 (host)) that have captured OVA-coated fluospheres in treated chimeric animals (n = 4). Data are means ± SEM. (Right) Representative per-cell uptake of fluospheres by macrophages of each haplotype. (**E**) Single-cell images of sorted skin macrophage subsets of a chimeric mouse (CD45.1/CD45.2, donor/host) injected i.v. with OVA-FITC fluospheres. Images were obtained using an imaging flow cytometer. Notably, host macrophages were mostly filled with melanin as confirmed with Fontana-Masson staining (*data not shown*).

The following figure supplement is available for figure 8:

**Figure supplement 1.** Comparison of OVA-FITC fluosphere uptake either by skin dextran+ macrophages in control, γ-irradiated and splenectomized animals or by spleen macrophages and skin macrophage subsets in chimeric mice.

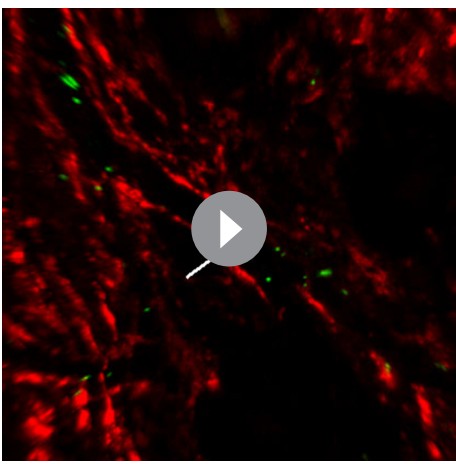

**Video 8.** In vivo uptake of blood-borne OVA-FITC fluospheres by STREAMs, example 1. A C57BL/6 mouse was first injected i.v. with HMw TRITC-dextran and with OVA-FITC fluospheres 16 hr later. Imaging began 24 hr after the last injection. The frames in the video sequence correspond to a z-stack (4 sections acquired every 10 μm) obtained every 3.300 s over a total period of 16 min. Interestingly, an intravascular STREAM's protrusion is detected at the beginning of the video sequence (white arrow). The video sequence shows free-flowing FITC-fluospheres as well as the intravascular cells that have phagocytosed them (most probably monocytes). Some of the FITC⁺ cells transiently interact with endothelium at sites coinciding with STREAM's intravascular protrusions. Intraluminal FITC-fluospheres retained in such intravascular protrusions can be also observed. Furthermore, FITC-fluospheres that have been previously phagocytosed by dextran⁺ STREAMs are detected in the extravascular tissue.

the $Ccr2^{-/-}/Ccr2^{+/+}$ (donor/host) chimeras (whose skin was defective in radio-sensitive macrophages, the repopulation from BM was prevented but the pool of STREAMs was intact) compared to that of control $Ccr2^{+/+}/Ccr2^{+/+}$ chimeras, using all parameters assayed previously (*Figure 10—figure supplement 1F–G*). Altogether these results strengthen the idea of the selective role for STREAMs in tissue repair.

## Discussion

In this study we have identified a unique subset of dermal perivascular macrophages, skin trans-endothelial radio-resistant anti-inflammatory macrophages (STREAM), which capture blood-borne macromolecules by extending protrusions into the vascular lumen at steady-state. The observation of this phenomenon in chimeric animals allowed us to further characterize STREAMs as a radio-resistant subset of skin macrophages, whose turnover is ensured by an extramedullary source. We could also determine that STREAMs are committed to perform anti-inflammatory functions, lacking the plasticity to be reprogrammed in the presence of a potent pro-inflammatory stimulus. Finally, the selective depletion of STREAMs highlighted their function in tissue repair and remodeling processes to regain skin homeostasis.

It is well established that specialized macrophage subtypes, such as subcapsular sinus macrophages in lymph nodes and Kupffer cells in liver, can traverse fenestrated endothelial sinusoids to capture lymph- or blood-borne particulate antigens and participate in the initiation of innate and adaptive immunity (*Huang et al., 2013*; *Junt et al., 2007*; *Lee et al., 2010*; *Wong et al., 2013*). Dendritic cells are also able to extend transepithelial projections to capture foreign particulate antigens from airways and gut mucosa, or traverse the vasculature in pancreatic islets of Langerhans (*Calderon et al., 2008*; *Chang et al., 2013*; *Farache et al., 2013*; *Hammad et al., 2009*; *Rescigno et al., 2001*). However, some of these specialized dendritic cells have been redefined as macrophages (*Calderon et al., 2014*; *Mazzini et al., 2014*). CX3CR1⁺ myeloid cells in the central nervous system could also gain access to vasculature (*Barkauskas et al., 2013*) and mast cells were recently reported to capture blood-borne IgE, albeit the underlying mechanism remains unknown (*Cheng et al., 2013*). Herein, we report that STREAMs are unique macrophages in that they traverse the endothelial junctions of non-fenestrated dermal vasculature to take up circulating macromolecules under homeostatic conditions. STREAM's protrusions only emerge through the endothelial cell-cell junctions and do not use a transcellular route, as illustrated using a genetic approach (the VE-cad-α-cat transgenic mouse line, which display impregnable, highly stable adherens junctions). Moreover, unlike subcapsular sinus macrophages in lymph nodes, which act like fly paper retaining lymph-borne viral particles on their surface for subsequent transfer to B cells (*Junt et al., 2007*), STREAMs internalize particulate antigens, such as OVA-FITC Fluospheres, and might play a role in their cross-presentation to tissue-resident memory CD8⁺ T cells.

The existence of radio-resistant dendritic cells in the skin has been previously reported (*Bogunovic et al., 2006*; *Merad et al., 2002*), whereas the presence of radio-resistant dermal macrophages has remained elusive hitherto. This study reveals that STREAMs are a radio-resistant subset

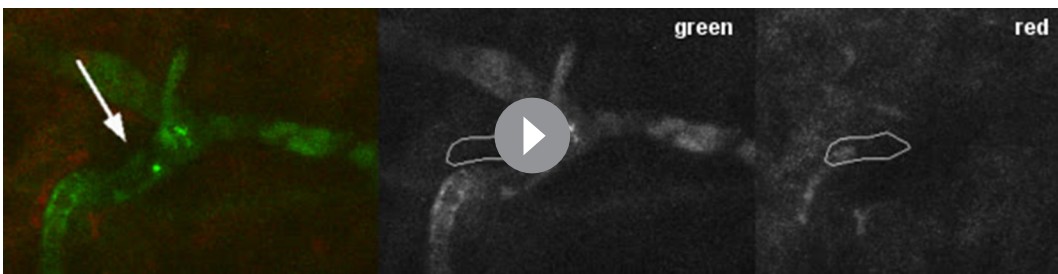

**Video 9.** In vivo uptake of blood-borne OVA-FITC fluospheres by STREAMs, example 2. The i.v. injection of HMw TRITC-dextran was carried out 16 hr prior to the experiment and the i.v. injection of OVA-FITC fluospheres preceded the beginning of imaging acquisition. The frames from the video sequence are maximal projections of a z-stack (4 sections acquired every 10 μm, 30 μm in depth) obtained every 2.27 sec over a total period of 11 min 30 s. The video shows a dextran[dim] STREAM that simultaneously phagocytoses several OVA-coated FITC fluospheres. The subsequent retrograde transport of phagosomes from the vessel towards the cell body is also shown. The white arrow in the merged image points to the place where fluospheres are captured from vasculature and the cell area is depicted in green and red channels for clarity.

that coexists in steady state with other skin macrophages of radio-sensitive nature. In this regard, the radio-resistant macrophages described in (*Haniffa et al., 2009*) might be the human homolog of this mouse subset, extending the generality and impact of our findings. The chimerism, parabiosis and fate mapping experiments also indicate that STREAMs are maintained locally in the skin throughout adult life independently of circulatory precursors, as opposed to radio-sensitive macrophages that are renewed at a low rate from blood in steady state and replaced by donor BM progenitors after a genotoxic insult. Thus, our data further extent the previous knowledge on the minimal blood exchange experienced by tissue-resident macrophages in other tissues (*Hashimoto et al., 2013*; *Yona et al., 2013*).

The renewal potential and the lifespan of tissue-resident macrophages are still controversial. Somes studies have described the potential of macrophages to be self-perpetuated under certain conditions (for example, in the presence of IL-4 during a parasitic infection [*Jenkins et al., 2011*] or in a M-CSF- or GM-CSF-dependent manner after macrophage depletion [*Hashimoto et al., 2013*]). STREAMs are unable to self-renew in steady-state and their turnover from a Bmi1[+] adult progenitor occurs at a low basal rate, becoming outcompeted over time by other skin-resident macrophages with faster turnover. Hence, our results concur with previous studies reporting a dual origin for skin macrophages with a prenatally-derived CCR2-independent subset and a postnatal BM-derived CCR2-dependent subset, being the prenatally-derived subset displaced by the BM-derived subset with age (*Jakubzick et al., 2013*; *Tamoutounour et al., 2013*). Interestingly, this dual origin has been also observed in other tissues such as heart (*Molawi et al., 2014*).

Unraveling the ontogenetic source for tissue-resident macrophages is another complex matter. There are several landmark studies describing the existence of different macrophage precursors during development. The yolk sac-derived Myb-independent F4/80[bright] macrophages seeded in the embryonic skin prior to birth and renewed independently of definitive hematopoiesis (*Schulz et al., 2012*) might correspond to the developmental precursor for STREAMs and the c-Myb dependent fetal liver monocytes that give rise to macrophages with self-renewal capacity (*Hoeffel et al., 2015*) might be the precursors of the radiosensitive macrophages (those susceptible of replenishment by BM precursors after a genotoxic insult). However, the elucidation of skin macrophage ontogeny deserves further thorough investigation in future studies.

Regarding phenotypic characterization, a classification of the skin mononuclear phagocyte system based on phenotypic markers has recently identified heterogeneity among the skin-resident macrophages, although not related to radio-sensitivity. These authors distinguish two distinct skin-resident macrophage subsets defined both as CD11b[+] CD11c[-] CD64[+] MerTK[+] Ly6C[low] CCR2[-] but differing on the expression of MHC-II (*Tamoutounour et al., 2013*). They suggest that both subsets possess analogous transcriptional profiles, share functions, are radio-sensitive and equally dependent on BM supply, and could represent sequential stages of differentiation (the MHC-II[low] subset (P4) maturing into MHC-II[high] (P5) macrophages). As demonstrated in *Figure 6A* and *Figure 6—figure*

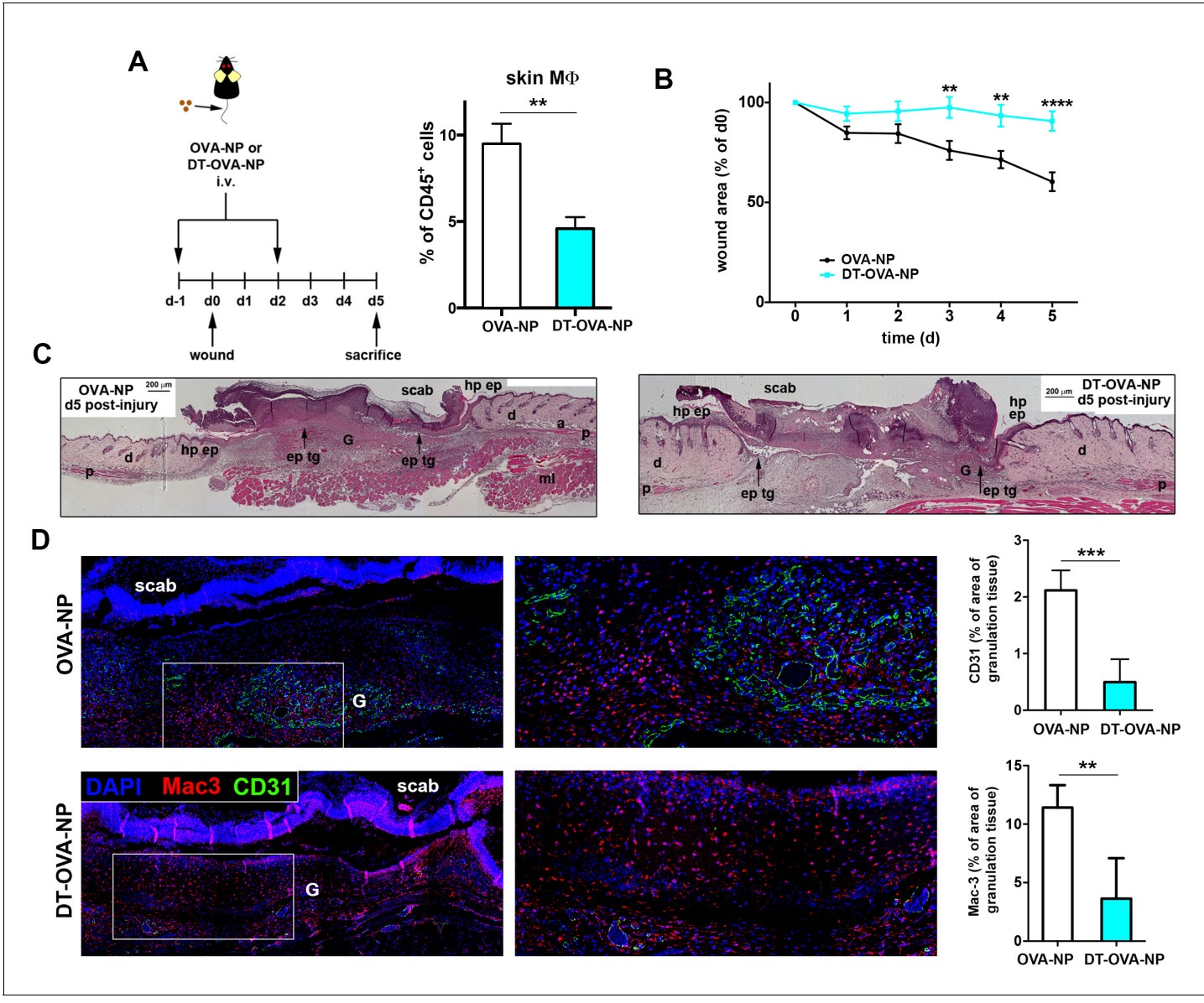

**Figure 9.** Macrophage depletion using DT-OVA-NP hampered wound healing. (**A**) (Left) Animals were first injected i.v. with either OVA-NP or DT-OVA-NP (2.5 μl/g) and, 1d later, the wound healing assay began. Two days after wounding, animals were treated with another similar dose of OVA-NP (control) or DT-OVA-NP. Then, mice were sacrificed 3d later (mid-stage repair) and their ears were collected to analyze skin macrophages. (Right) Bar histogram showing the percentage of MΦs out of total CD45[+] cells in the ears of OVA-NP- or DT-OVA-NP-treated animals. Data are mean ± SEM (n = 4–7). Statistical significance was assessed by unpaired two-tailed Student's t-test (**p-value < 0.01). (**B**) Analysis of mid-stage repair (5d after wounding). The plot illustrates the decrease of the wound area over time expressed as percentage of the initial wound. Data are mean SEM, n = 32 wounds/group. Statistical significance was assessed by two-way ANOVA analysis with Sidak's post-test (**p-value < 0.01, ****p-value < 0.0001) (**C**) Histological analysis of wounds at mid-stage of repair from OVA-NP- and DT-OVA-NP-treated animals. Hematoxylin-eosin staining of representative samples is shown. (Hp ep: hyperproliferative epithelium, ep tg: epithelial tongue, G: granulation tissue, d: dermis, p: panniculus carnosus, a: adipose tissue, ml: muscle layer). (**D**) (Left) Representative immunofluorescence staining of vessels (CD31 staining) and macrophages (Mac-3 staining) in tissue sections of wounds at mid-stage (5d) from OVA-NP- or DT-OVA-NP-treated animals. (Right) Quantification of CD31 and Mac-3 stained area within the granulation tissue is shown. Data are mean ± SD (n = 4–6). Statistical significance was assessed by unpaired two-tailed Student's t-test (**p<0.01, ***p<0.005).

The following figure supplement is available for figure 9:

**Figure supplement 1.** Selective deletion of STREAMs after DT-OVA-NP treatment.

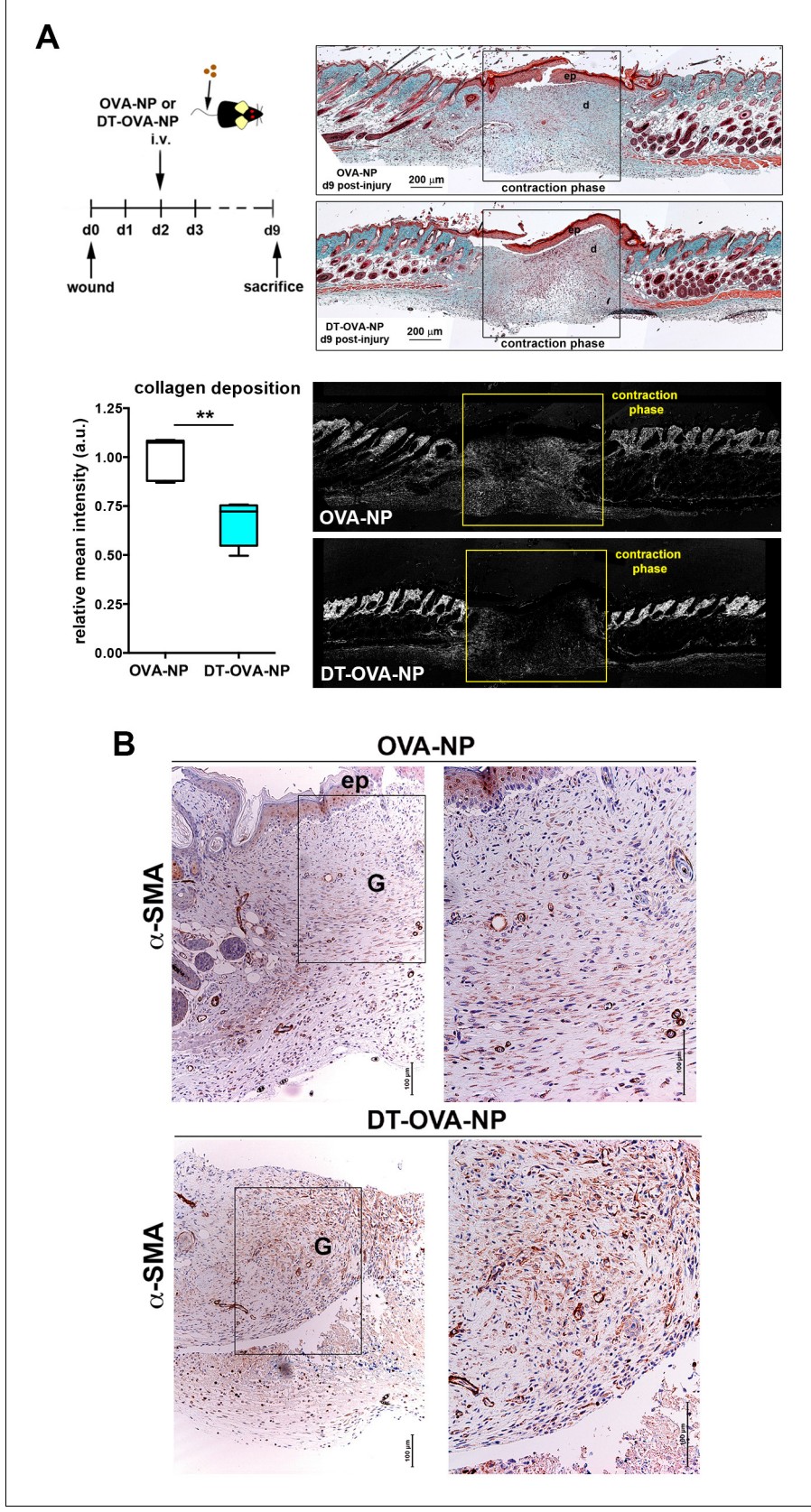

**Figure 10.** Macrophage depletion using DT-OVA-NP impaired collagen deposition and myofibroblast organization within the wound. (A) Wound healing assay until closure with administration of DT-OVA-NP or OVA-

*Figure 10 continued on next page*

*Figure 10 continued*

NP 2d after injury. After sacrifice, tissue was stained with hematoxylin-eosin (upper right) or analyzed for its content in collagen using Masson trichrome staining (lower right) followed by relative intensity quantification (n = 6/group) (lower left). Statistical significance was assessed by an unpaired two-tailed Student's t-test (**p<0.01). (Ep: epidermis, d: dermis). (**B**) Analysis of the myofibroblast distribution in the wound area at the time of closure (contraction phase) in OVA-NP and DT-OVA-NP mice. Immunohistochemical staining of α-SMA was performed. (Left) View of the complete wound, (right) zoom in the granulation tissue.

The following figure supplement is available for figure 10:

**Figure supplement 1.** Skin-resident macrophages different from STREAMs are not critically involved in tissue repair.

*supplement 1B*, the expression of MHC-II in skin macrophages cannot precisely discriminate the two independent subsets of macrophages with distinct radio-sensitivity, different polarization profiles and non-overlapping functions described in our work. Therefore, the classification criteria used in this work allows better resolution of the skin macrophage subsets, compared to classifications relying on subset-defining markers that might not adequately reflect the existing heterogeneity within the skin niche.

Metabolic adaptation is a key feature in macrophage activation, instrumental for their function in homeostasis, immunity, and inflammation (*Martinez et al., 2013*). In this regard, the GSEA of our RNASeq data highlight clear differences in the metabolic gene profiles of host radio-resistant and BM-derived macrophages. We found an enrichment in genes involved in glycerolipid, glycerophospholipid and linoleic acid metabolism, as well as in glycosphingolipid biosynthesis in the gene profile of radio-resistant macrophages. These results are compatible with the energetic requirements and the regulation of membrane fluidity needed for phagocytosis and are also crucial for tissue remodeling (*Biswas and Mantovani, 2012*). Other genes enriched in this subset are related to fibrogenesis and tissue repair such as *Tgfb1* and genes of the Hedgehog signaling pathway, as well as genes related to taurine and hypotaurine metabolism, being consistent this with an antioxidant protective role. Conversely, the transcriptional profile of the BM-derived macrophages includes an over-representation of genes associated with the carbohydrate and retinol metabolisms such as the pentose phosphate pathway, with the ABC transporters and the JAK-STAT and WNT signaling pathways, in accordance with their pro-inflammatory potential. Moreover, a gene encoding a regulator of the metabolic switch from oxidative phosphorylation to aerobic glycolysis (*Trap1*) (*Yoshida et al., 2013*) is specifically expressed in BM-derived macrophages.

Our study in the homeostatic skin along with others describing the existence of polarized macrophages already in steady state in heart, testis and pancreas (*Calderon et al., 2015*; *DeFalco et al., 2014*; *Pinto et al., 2012*) help to reformulate the long-held paradigm of macrophage polarization, which defined the functional specialization of macrophages from a steady-state non polarized status mainly as a consequence of integrating external stimuli (*Gordon and Pluddemann, 2013*; *Sica and Mantovani, 2012*). Interestingly, the study by Calderon and colleagues (*Calderon et al., 2015*) highlights the imprinting of distinct features in macrophages by their anatomical localization within the steady-state pancreas. The macrophages in the islets possess a pro-inflammatory M1 phenotype and protrude into the fenestrated vessels, whereas the stromal milieu sets the response of the macrophages in favor of the M2 phenotype. This pattern of regional specialization might not be applicable to the skin where, e.g., perivascular macrophages with pro-inflammatory (*Abtin et al., 2014*) and anti-inflammatory (STREAMs) phenotypes coexist in the same areas. However, although both subsets mostly share environmental stimuli, STREAMs can get access to the intravascular milieu and might receive selective signals that could contribute to maintain their phenotype. Therefore, the essential role of the environment in controlling tissue-specific macrophage identities cannot be dismissed as highlighted in recent epigenetics studies (*Gosselin et al., 2014*; *Lavin et al., 2014*).

The preferential depletion of anti-inflammatory STREAMs has a deleterious impact in the wounded skin, indicating that they could participate in orchestrating the pro-reparative functions traditionally associated to the so-called alternatively activated (M2) macrophages (*Das et al., 2015*). Conversely, we observed that the rate of wound closure was neither affected in the *Ccr2*[-/-] animals

(as previously reported by *Willenborg et al., 2012*) nor in chimeric *Ccr2*$^{-/-}$/*Ccr2*$^{+/+}$ mice, indicating that infiltrating monocytes and radiosensitive tissue-resident macrophages could be mostly dispensable for the reparative phase of the healing process in the skin. Thus, the coexistence of macrophage subsets performing non-redundant roles in homeostatic skin highlights the need of maintaining equilibrium between both pools to preserve the skin fitness. The lack of functional redundancy could be of clinical significance, and may contribute to better understand concomitant skin pathologies arising from radio-induced bone marrow ablation therapies and certain inflammatory skin disorders, particularly those aggravated with age.

## Materials and methods

### Mice

Male either littermates or age-matched (8-to-12-week-old) mice on the C57BL/6J background were used for intravital imaging, phenotypic characterization, transcriptional analysis and functional experiments. Chimeric animals were generated as follows: 8-to-12-week-old C57BL/6 wt male mice (CD45.2 haplotype), were lethally γ-irradiated with 2 doses of 6.5 Gy and transplanted with a mixture of 5 x 10$^6$ BM cells from either B6 SJL mice (CD45.1 haplotype), or B6 ACTB-eGFP mice. Alternatively, irradiated B6 SJL mice (CD45.1$^+$) were reconstituted with BM from *Ccr2*$^{+/+}$ or *Ccr2*$^{-/-}$ (CD45.2$^+$) mice. Animals were used for experimental procedures after 12 to 18 weeks of reconstitution. Details of the generation and characterization of the gene-targeted mouse expressing VE-cadherin-α-catenin protein can be found in (*Schulte et al., 2011*). Inducible *Bmi1*-IRES-Cre-ERT2 Rosa26 YFP mice were generated by crossing the *Bmi1*$^{CreER/+}$ strain (*Sangiorgi and Capecchi, 2008*) with Rosa26$^{YFP/+}$ reporter mice. *Bmi1*-IRES-Cre-ERT2 Rosa26 YFP double heterozygous mice were injected with tamoxifen 5d prior to analysis. Tamoxifen (Sigma-Aldrich [St Louis, MO]) was dissolved in corn oil (Sigma-Aldrich) to a final concentration of 20 mg/ml, and mice were i.p. injected every 24 hr on three consecutive days with tamoxifen at a concentration of 9 mg per 40 g body weight. The nestin-eGFP reporter strain was a kind gift from Dr. G. Enikolopov (*Mignone et al., 2004*). DPE-GFP mice were generated as described in (*Mempel et al., 2006*). *Langerin*-eGFP, *Cx3cr1*-GFP, *Lyz2*-Cre:Rosa26YFP and *Ccr2*$^{-/-}$ mice were kindly provided by Dr. B. Malissen, Dr A. Hidalgo, Dr. M. Ricote and Dr. C. Ardavin, respectively. Mice were kept in pathogen-free conditions in the CNIC Animal Unit, Madrid. Animal studies were approved by the local ethics committee and by the Division of Animal Protection of Comunidad de Madrid (approved protocols PROEX 159/15 and 160/15). All animal procedures conformed to EU Directive 2010/63EU and Recommendation 2007/526/EC regarding the protection of animals used for experimental and other scientific purposes, enforced in Spanish law under Real Decreto 1201/2005.

### Intravital imaging

Ears were taped to the center of a coverslip and attached with high vacuum grease. Hairless areas were examined using a HCX PL APO lambda blue 20.0X 0.70 IMM UV objective (multi-immersion, glycerol) coupled to an inverted microscope (DMI6000; Leica Microsystems GmbH [Wetzlar, Germany]) equipped with a confocal laser-scanning unit (TCS-SP5; Leica). Non-invasive intravital imaging procedures were carried out in a thermostatic chamber at 37°C. For short-term studies (1–2 hr), animals were first anesthetized by i.p. injection of ketamine (50 mg/kg), xylazine (10 mg/kg), and acepromazine (1.7 mg/kg), and repeated half-doses were administered when needed over the course of the experiments. For long-term studies (time-lapse imaging > 2 hr), animals were first anesthetized by i.p. injection of urethane (1.2 gr/kg) dissolved in PBS, followed by s.c. injection of 1/10 of the initial dose dissolved in 200 μl PBS for rehydration after several hours. As vascular tracers, 100 μl of 2 MDa FITC- or TRITC-dextran (1% w/v, which corresponds to a concentration of 5 μM; anionic lysine fixable [Molecular Probes, Thermo Fisher Scientific, Waltham, MA]), or 25 μg Vivotag (VisEn Medical, PerkinElmer [Waltham, MA]) dissolved in PBS were i.v injected. For in vivo staining of the vasculature, an anti-CD31 antibody (Fitzgerald Industries [North Acton, MA]) was labeled in house with an Alexa Fluor 647 monoclonal antibody labeling kit (Molecular Probes), and 25 μg were injected i.v. 16 hr after dextran administration. Images were obtained using bidirectional scanning mode and the acquisition of the different channels was sequential to avoid fluorescence bleed-through artifacts. Z-stack images were acquired with optimal confocal sectioning (spaced 0.63 μm along the z-axis).

For time-lapse acquisition, z-sections were obtained at 10 µm intervals up to a depth of 70–100 µm, using a pinhole aperture > airy 1 (confocal acquisition). The acquisition rate was ~2–3 sec/z-stack.

## Whole-mount staining of skin samples

Live mice anesthetized with a mixture of zoletil and dontor were surgically opened and perfused with 1% paraformaldehyde (PFA) in PBS at a continuous infusion rate of 7 ml/min, using a programmable syringe (Harvard Aparatus). Different parts of the skin were collected and post-fixed in 1% PFA for 1 hr at room temperature (RT). Tissues were stained as described in (Baluk et al., 2007). Briefly, tissues were incubated overnight (O/N) at RT with PBS containing 0.3% Triton X-100, 5% goat serum (Jackson ImmunoResearch Europe Ltd. [Suffolk, UK]) and primary antibodies. The anti-murine primary antibodies used were hamster anti-CD31 clone 2H8 (Chemicon, EMD Millipore [Billerica, MA]), rat anti-CD68 clone FA-11 and polyclonal rabbit anti-collagen IV (Abd Serotec, Bio-rad [Hercules, CA]), polyclonal rabbit anti-CD31 and polyclonal chicken anti-GFP (Abcam [Cambridge, UK]), polyclonal rabbit anti-LYVE-1 (ReliaTech [Wolfenbüttel, Germany]), and biotin-conjugated CD45.2 clone 104 (BD Pharmingen, BD Biosciences [San Diego, CA]). Samples were then thoroughly washed with 0.3% Triton X-100 in PBS and stained with appropriate secondary antibodies O/N at RT. Secondary antibodies used were DyLight 405 donkey anti-rat, Cy3 donkey anti-chicken and Alexa Fluor 647 or 488 goat anti-armenian hamster (Jackson Immunoresearch), Rhodamine-X goat anti-rabbit, Alexa 647 goat anti-rabbit, Alexa 647 chicken-anti-rat, Alexa 488 donkey-anti-rat and Alexa 647 or 488 goat-anti-mouse (Molecular Probes). After the last round of washing, labeled samples were fixed in 1% PFA for 30 min and mounted for microscopy analysis in Prolong Gold antifade reagent (Molecular Probes). Confocal z-stacks up to a depth of 100 µm (including epidermis and upper dermis) were obtained using a LSM 700 laser scanning microscope equipped with a LD LCI Plan/Apochromat 25x/0.8 Imm Korr DIC M27 objective (Carl Zeiss AG [Oberkochen, Germany]).

## Image processing and analysis

Confocal z-stack images from fixed samples or intravital experiments were processed to obtain maximal projections, orthogonal sectionings, 3D reconstructions or isosurface renderings, fluorescence intensity profiles, colocalization analyses and 3D computational analyses of distances using Imaris 7.3.1 (Bitplane [Belfast, UK]), Volocity 5.5.1 (PerkinElmer), and ImageJ 1.42q (NIH, [Bethesda, MD]). Images from in vivo time-lapse acquisitions were analyzed and videosequences set up and edited either with Imaris or Volocity softwares.

## Dextran 'pulse-chase' assay

Animals were injected i.v. with 100 µl 1% HMw TRITC-dextran and, after a lapse of 24 hr, with 100 µl 1% HMw FITC-dextran. Animals were sacrificed 24 hr after the last injection and ears were processed for imaging.

## In vivo staining of intraluminal protrusions of macrophages

Mice were injected retro-orbitally with 10 µg of Alexa 488 anti-CD206 (clone C068C2, BioLegend [San Diego, CA]) or Alexa 488 control isotype antibody (rat IgG2b, κ) and sacrificed 3 min later. Then, animals were perfused with PBS and ear skin was processed for flow cytometry. Counter-staining of CD206 was performed with Alexa 647 anti-CD206 clone MR5D3 (BioLegend).

## Flow cytometry analysis and sorting strategies

Skin samples (ears and hind footpads) were digested either with 2 mg/ml crude collagenase type IA (Gibco, Thermo Fisher Scientific [Waltham, MA]) in PBS for 2 hr or with 0.25 µg/ml Liberase TM Research Grade (Roche Holding AG [Basel, Switzerland]) in RPMI medium for 1 hr at 37°C. Samples were then mechanically disrupted, washed and filtered. Single-cell suspensions were incubated with anti-mouse FcRII/III (clone 2.4G2) for 10 min at 4°C in PBS containing 0.05% BSA and 0.05 mM EDTA, and then stained with the following antibodies: Pecy7 anti-F4/80 clone BM8, APC or APC-eFluor 780 anti-CD45.2 clone 104, PE anti-CD115 clone AFS98, PE anti-Tie-2 clone Tek4, and APC-eFluor 780 Streptavidin (eBioscience [San Diego, CA]); v450 Horizon anti-CD45 Clone 30-F11, v450 Horizon or PerCP-Cy5.5 anti-CD45.1 clone A.20, PE anti-CD11c clone HL3, FITC or APC or APCCy7

anti-MHC-II clone M5/114.15.2, APC anti-MHC-I clone AF6-88.5, PE anti-Ly6C clone HK1.4, PECy7 or Alexa Fluor 647 anti-CD64 clone X54-5/7.1, PE anti-CD70 clone FR70, PE anti-CD86 clone GL-1, FITC anti-CD40 clone 3/23, and FITC anti-CD11b clone M1/70 (BD Pharmingen); APC anti-CCR2 and biotinylated MertK (R&D Systems Inc. [Minneapolis, MN]); Streptavidin Alexa Fluor 350 (Molecular Probes); APC or PerCP-Cy5.5 anti-CD68 clone FA-11, and Alexa Fluor 647 CD326 clone G8.8 (BioLegend); Alexa Fluor 647 anti-CD206 (AbD serotec); and rabbit polyclonal anti-Tomm20 clone FL-145 (Santa Cruz Biotechnology [Dallas, Tx]). For intracellular staining, samples were first fixed and permeabilized with Fix & Perm before antibody incubations in Wash & Perm buffer (BD Biosciences [San Diego, CA]). For Ki-67 staining, samples were treated following manufacturer's protocol (Alexa Fluor 647 anti-Ki-67, BD Pharmingen). Single-cell suspensions were analyzed with a FACSCanto II HTS cytometer or a FACSAria II SORP sorter (BD Biosciences), and data were processed with FlowJo 7.6.3 and BD FACSDiva v.6.1.3. softwares. In some cases, sorted samples were further analyzed by combining flow cytometry and imaging with Flowsight (Amnis, EMD Millipore [Billerica, MA]).

## Parabiosis surgery

For parabiosis, gender- and weight-matched mice were anesthetized i.p. with a mixture of ketamine (100 mg/kg), xylazine (10 mg/kg), and acepromazine (3 mg/kg). Hair was removed from the lateral aspects of the mice by shaving and hair removal cream. A longitudinal skin incision was made from the olecranon to the knee joint on opposing sides of each mouse of the parabiotic pair. Animals were placed side-by-side and the right olecranon of one animal was attached to the left olecranon of the other by a double suture. The equivalent procedure was performed for the knee joints to further secure the parabiotic pair. The dorsal and ventral skins were then approximated continuous 6–0 suture and by staples along the lateral abdominal aspect of the mice. Parabiosis was maintained for 6 months and, then, one of the partners in the parabiotic pair was injected retro-orbitally with 200 µl of HMw TRITC-dextran (5 µM) and animals were sacrificed 3d later. Chimerism in skin-resident macrophages was determined as well as the capture of intraluminal dextran by host and partner macrophages.

## Solar-simulated UV radiation

Animals were exposed to UV irradiation using a 1000 watt xenon arc solar simulator (Oriel by Newport [Irvine, CA] ) equipped with an Oriel 81,017 filter (Colipa [Oudergem, Belgium]). UVB and UVA irradiance measurements were performed using an IL-1700 radiometer (International Light Technologies Inc. [Peabody, MA]) equipped with SED240/UVB-1/TD and SED033/UVA/TD photodetectors. The radiometer was calibrated with a Solar-Scope spectroradiometer (Solatell [Croydon, UK]). Animals were irradiated with 5 J/cm$^2$ UVA-UVB except for their left ear that was covered to be used as non-treated control. Then, they were allowed to recover for 1 month before sacrifice.

## Clodronate liposome-mediated depletion of skin macrophages

Clodronate (Roche) or PBS liposomes were injected s.c. (20 µl/hind footpad). Skin samples were processed for flow cytometry analysis after 24 hr or 7d.

## In vivo BrdU treatment

Mice were allowed to drink BrdU-treated water (0.8 mg/ml) for 8d and then were sacrificed. BrdU incorporation in skin cell subsets was analyzed by flow cytometry (FITC BrdU Flow Kit, BD Pharmingen).

## Measurement of mitochondria function

Ex vivo staining of mitochondria activity was conducted by flow cytometry following a previous protocol (*Johnson and Rabinovitch, 2012*). Briefly, ears were excised and split in half. The tissue was immediately incubated in the staining solution (20 nM of MitoTracker Orange CMTMRos [Molecular Probes] diluted in HBSS1X with 5% FBS) for 20 min at 37°C, 5% CO2. Then tissues were digested and dye was maintained at 10 nM in all the following steps before FACS analysis.

## RNA-Seq library production

cDNA was synthesized and amplified directly from cells using the Smarter Ultra Low RNA kit (Clontech Laboratories Inc. [Mountain View, CA]). Amplified cDNAs (10 ng) were fragmented using Covaris E220 (Covaris [Woburn, MA]) to an average fragment size of approximately 150 pb. Index-tagged sequencing libraries were generated from the fragmented cDNAs using the TruSeq RNA Sample Preparation v2 Kit (Illumina Inc. [San Diego, CA]), starting from the End Repair step. Libraries were quantified using a Nanodrop spectrophotometer (Thermo Fisher Scientific [Waltham, MA]) and their size distributions were determined using the Bioanalyzer DNA-1000 Kit (Agilent). Libraries were sequenced on the Genome Analyzer IIx (Illumina) following the standard sequencing protocol with the TruSeq SBS Kit v5 (Illumina). Fastq files containing the sequencing reads for each library were extracted and demultiplexed using Casava v1.8.2 (Illumina).

## RNA-Seq analysis

Reads were pre-processed with Cutadapt (*Martin, 2011*), to remove both TruSeq adaptor and SMARTer primer sequences. The resulting reads were mapped on the mouse transcriptome (Ensembl gene-build GRCm38.v70) and genome, using RSEM v1.2.3 (*Li and Dewey, 2011*) and Bowtie2 v2.0.6 (*Langmead and Salzberg, 2012*), respectively. The observation that a significant fraction of reads could be mapped on the genome, but not on the transcriptome, in close proximity to dA/dT-rich sequences, suggested that traces of contaminant genomic DNA could have been artifactually amplified by poly (A) priming. To avoid overestimation of transcript read counts, read clusters mapped on the genome at less than 50 bp downstream of dT-rich or upstream dA-rich regions were discarded. Filtered reads were mapped again on the transcriptome, and quantified, with RSEM. Only genes with at least 2 counts per million in at least 2 samples were considered for statistical analysis. Data were then normalized and differential expression tested using the Bioconductor package EdgeR (*Robinson et al., 2010*). Hierarchical Clustering was run using Genesis Software (*Sturn et al., 2002*) on the normalized expression profiles of the subset of genes with a p-value smaller than 0.05 and similar behavior in the two replicates of each cell type. Gene Set Enrichment Analysis (GSEA) (*Subramanian et al., 2005*) was run on the list of genes expressed in at least one cell type. The RNASeq dataset is available at http://www.ncbi.nlm.nih.gov/geo/query/acc.cgi?token=gzijkssyjraldqd&acc=GSE50566 (authors: Francisco Sanchez-Madrid and Olga Barreiro; year of publication: Sep 4th 2013; title: Homeostatic skin contains two different subsets of resident macrophages with distinct origin and gene profile; GEO Accession Number GSE50566).

## RNA extraction and immune gene profile analysis

RNA was extracted from sorted macrophage populations obtained from chimeric CD45.1 (donor)-CD45.2 (host) animals using the Absolutely RNA Nanoprep Kit (Agilent Technologies [Santa Clara, CA]). RNA quantity and quality were determined using a 2100 Bioanalyzer (Agilent Technologies) and a Nanodrop-1000 Spectrophotometer (Thermo Fisher Scientific). Each RNA sample was amplified using the MessageAmp II aRNA Amplification Kit (Ambion). Total amplified mRNA (30 ng) was converted to cDNA and loaded on a TaqMan Array Micro Fluidic Card (Applied Biosystems, Life Technologies, Thermo Fisher Scientific [Foster City, CA]). Relative gene expression was calculated with the ABI Prism 7900 HT Sequence Detection System (Applied Biosystems) and Qbase software (Biogazelle [Gent, Belgium]), using *Actb*, *B2m*, *Hprt1*, *Gusb* and *Gapdh* genes as reference targets for the analysis of the immune system and inflammation gene signature array and *18S*, *Raf1*, *Ctnnb1*, *Eef1a1* for the analysis of the stem cell gene signature array. A comparative gene expression analysis was carried out by calculating ΔlogCt from the average logCt for every gene from each type of tissue-resident macrophages.

## Analysis of cytokine secretion by FACs and ELISA

Sorted skin anti- and pro-inflammatory macrophage populations were obtained from the skin of C57BL6 mice. Cells were seeded (40,000 cells/100 µl RPMI medium) and stimulated in vitro with LPS (1 µg/ml) for 24 hr. Supernatants were used for analysis of cytokine secretion using Mouse Inflammation (17-Plex) multiplex kit (ANTIGENIX AMERICA Inc [Huntington Station, NY]). TGF-β and IL-10 secretion were analyzed with an ELISA kit (eBioscience).

## Cytospin and staining

Sorted donor and host skin macrophages of quimeric mice were diluted in PBS containing 0.5% BSA and centrifuged in a cytospin centrifuge for 10 min at 800 rpm. Samples were stained with rabbit polyclonal anti-arginase I clone H-52 (Santa Cruz Biotechnology) or rabbit polyclonal anti-heme oxygenase-1 (Chemicon) following a standard immunofluorescence protocol. Then, samples were analyzed in the LSM 700 laser scanning microscope described above.

## In vivo uptake of particulate OVA

FITC fluospheres (Ø 0.2 µm) (Fluospheres, sulfate microspheres, Molecular Probes) were diluted in PBS to a final concentration of 0.5% solids and adsorbed with 1 mg/ml endotoxin-free OVA (Calbiochem, EMD Millipore [Billerica, MA]) following the manufacturer's instructions. Then, 200 µl of the colloidal dissolution were injected i.v. in either untreated, lethally γ-irradiated or splenectomized animals. Seventy-two hours later, animals were sacrificed and manually perfused with PBS to eliminate remaining fluospheres from the vasculature. In some cases, hind footpads were treated with clodronate or PBS liposomes 72 hr before the injection of fluospheres. Clodronate or PBS liposomes (ClodronateLiposomes.org) were injected s.c. (20 µl/hind footpad). For splenectomy, animals were anesthetized with a mixture of ketamine/xylazine (100/10 mg/Kg), the left lateral side was depilated and a small incision was made to remove the organ after ligation of the splenic vessels. Then, animals were allowed to recover from surgery for 1 month before proceeding with experiments.

## Selective depletion of STREAMs in skin

Home-made ultrasmall superparamagnetic iron oxide nanoparticles (*Groult et al., 2015*) (abbreviated as NPs, 1 ml, 3 mg Fe/ml) with 10 nm of core, 40 nm of hydrodynamic size and −35 mV of ζ-potential were incubated with endotoxin-free OVA (500 µl, 10 mg/ml) for 1 hr at R.T. Non-adsorbed OVA remnants were washed out and further incubation of OVA-NP with diphtheria toxin (DT) (50 µl, 1 mg/ml) (3 hr at R.T.) were carried out, followed by extensive washing. Then, mice were injected i.v. with 1 dose (2.5 µl/g) or 2 doses (spaced 3d apart) of this colloidal suspension and skin samples were analyzed by FACS and immunohistochemistry at different time-points.

## Wound healing assay

The back of mice was depilated and left to recover homeostasis (24 hr). Then, animals were anesthetized with isoflurane by inhalation. Four skin excisions of 5 mm in diameter were carried out using a bio-punch (Kai Industries Co., ltd [Seki, Japan]). Wounds were routinely measured every 24 hr until the sacrifice of the animals. Animals were injected i.v. either DT-OVA-NP or OVA-NP before or during the wound healing assay. After sacrifice, wounded tissue was embedded in paraffin and processed for histological analysis.

## Histological analysis

Standard Hematoxylin-Eosin and Masson trichrome stainings were performed in wound sections. Alternatively, samples were processed for standard immunohistochemistry staining using anti-αSMA (clone 1A4, Sigma-Aldrich) or for immunofluorescence using anti-Mac-3 (clone M3/84, Santa Cruz Biotechnology), anti-CD31 (rabbit polyclonal anti-mouse/human, Abcam), and DAPI. Stained samples were analyzed on a Leica DM2500 microscope coupled to a Leica DFC420 camera using 5x, 10x and 20x objectives. Intensity quantification of collagen deposition and quantification of macrophages and vessels in the granulation tissue were performed using ImageJ 1.42q.

## Statistical analysis

Normality of data distribution was assessed with the Kolmogorov-Smirnov test. Statistical significance was calculated either with an unpaired two-tailed Student's t-test, one-sample t-test, one-way ANOVA followed by either Tukey's or Dunnett's post-tests or two-way ANOVA followed by Sidak's or Bonferroni post-test as indicated. All statistical analyses were carried out with GraphPad Prism v5 (GraphPad Software Inc. [La Jolla, CA]).

## Acknowledgements

We thank Dr. M Vicente-Manzanares and S Bartlett for English editing, and Dr. I Mazo, Dr. Hortensia de la Fuente, Ana Domínguez and Raquel Sánchez for technical assistance. We also thank the Microscopy, Cellomics, Genomics, Bioinformatics, Advanced Imaging and Comparative Medicine Units at CNIC and the HMS Center for Immune Imaging for technical support. This work was funded by grants from the Spanish Ministry of Economy and Competitiveness (SAF2011-27330 to PM; and SAF2011-25834 and SAF 2014-55579-R to FS-M); grant INDISNET-S2011/BMD-2332 from the Comunidad de Madrid to PM and FS-M; Red Cardiovascular RD 12-0042-0056 from Instituto Salud Carlos III (ISCIII), and ERC-2011-AdG294340-GENTRIS to FS-M; and National Institutes of Health grant RO1 AR068383 to UHvA. OB is supported by a fellowship from Fundacion Alfonso Martin Escudero, ERC and RO1 grants. DC is supported by SAF grant. The CNIC is supported by the Spanish Ministry of Economy and Competitiveness and the Pro CNIC Foundation.

## Additional information

### Funding

| Funder | Grant reference number | Author |
| --- | --- | --- |
| Fundación Alfonso Martín Escudero | | Olga Barreiro |
| European Research Council | ERC-2011-AdG294340-GENTRIS | Olga Barreiro<br>Francisco Sánchez Madrid |
| National Institutes of Health | RO1 AR068383 | Olga Barreiro<br>Ulrich H von Andrian |
| Ministerio de Economia y Competitividad España | SAF2011-25834 | Danay Cibrian<br>Francisco Sánchez Madrid |
| Ministerio de Economia y Competitividad España | SAF 2014-55579-R | Francisco Sánchez Madrid<br>Danay Cibrian |
| Ministerio de Economia y Competitividad España | SAF2011-27330 | Pilar Martín |
| Comunidad de Madrid | INDISNET-S2011/BMD-2332 | Pilar Martín<br>Francisco Sánchez Madrid |
| Instituto Salud Carlos III | Red Cardiovascular RD 12-0042-0056 | Francisco Sánchez Madrid |

The funders had no role in study design, data collection and interpretation, or the decision to submit the work for publication.

### Author contributions

OB, Designed research plan, Performed experiments, Analyzed and interpreted data, Wrote the manuscript; DC, Performed experiments, Collaborated in experimental design, Analysis and interpretation of data, ; CC, Participated in wound healing experiments, Acquisition of data, Drafting or revising the article; DA, Performed parabiosis surgery, Acquisition of data, Drafting or revising the article; VM, Provided technical expertise for whole-mount staining, Drafting or revising the article, Contributed unpublished essential data or reagents; ÍV, AB, DV, Contributed with reagents, Drafting or revising the article; AGA, Helped in designing wound healing experiments, Drafting or revising the article; PM, Helped to plan research and collaborated in some experimental approaches and data analyses, Drafting or revising the article; UHvA, Contributed with reagents, Discussed results, Helped with manuscript writing; FSM, Planned research, Discussed results, Wrote the manuscript

### Author ORCIDs

Pilar Martín, http://orcid.org/0000-0002-2392-1764
Francisco Sánchez Madrid, http://orcid.org/0000-0001-5303-0762

## Ethics

Animal experimentation: Animal studies were approved by the local ethics committee and by the Division of Animal Protection of Comunidad de Madrid (approved protocols PROEX 159/15 and 160/15). All animal procedures conformed to EU Directive 2010/63EU and Recommendation 2007/526/EC regarding the protection of animals used for experimental and other scientific purposes, enforced in Spanish law under Real Decreto 1201/2005

# Additional files

## Supplementary files

• Supplementary file 1. List of genes of the RNA-Seq analysis differentially express in host versus donor macrophages in chimeric mice. Those protein coding sequences showing similar normalized expression levels across replicates are highlighted in bold.

• Supplementary file 2. Comprehensive Gene Set Enrichment Analysis (GSEA) of all genes expressed in at least one cell type comparing Kegg pathways upregulated in host (first datasheet) versus donor macrophages (second datasheet) of chimeric mice. Kegg pathways upregulated at a nominal p-value less than 1% appeared in bold. Among them, those categories with a FDR q-value proximal to 25% are highlighted in grey.

## Major datasets

The following dataset was generated:

| Author(s) | Year | Dataset title | Dataset URL | Database, license, and accessibility information |
|---|---|---|---|---|
| Francisco Sánchez Madrid, Olga Barreiro | 2013 | Homeostatic skin contains two different subsets of resident macrophages with distinct origin and gene profile | http://www.ncbi.nlm.nih.gov/geo/query/acc.cgi?acc=GSE50566 | Publicly available at the NCBI Gene Expression Omnibus (Accession no: GSE50556) |

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
