## [Decision Letter]

Thank you for submitting your article "Pivotal role for Skin Trans-endothelial Radio-resistant Anti-inflammatory Macrophages in tissue repair" for consideration by *eLife*. Your article has been favorably evaluated by Tadatsugu Taniguchi as the Senior editor and three reviewers, including Klaus Ley and Satyajit Rath, who is a member of our Board of Reviewing Editors.

The reviewers have discussed the reviews with one another and the Reviewing Editor has drafted this decision to help you prepare a revised submission.

Summary:

The authors convincingly and elegantly show a new subset of perivascular skin macrophages ("STREAM") that are radio-resistant, take up high molecular weight dextran by intraluminal processes, have a gene expression profile consistent with anti-inflammatory functions, and show functional importance in wound healing. The combination of lineage origin, anatomical location and specific functionality makes this an interesting set of findings of broad interest.

It would help to have the following concerns addressed.

Essential revisions:

1) It would be very useful to have data and clarity about the developmental origin of these cells, and the relationship of this macrophage group with other ontogenetically distinct subsets of dermal macrophages previously reported. Dual origins for skin macrophages have been previously reported as a CCR2-independent prenatally derived subset and a CCR2-dependent postnatal bone marrow-derived subset (Jakubzick et al. and Tamoutounour et al. 2013). Is the origin of these STREAM macrophages yolk-sac and/or fetal liver? Is their development c-Myb-dependent? Are these cells the same as or different from the perivascular macrophages that mediate neutrophil chemo-attraction (Abtin et al.)?

2) The authors refer to this macrophage subset as 'radio-resistant' and suggest that these cells are maintained locally. These are two separate properties though perhaps related, and require elucidation. Thus, does 'radio-resistance' refer to the differentiated perivascular cells themselves, or to some progenitor population? Can comparable numbers of STREAM cells be detected in perivascular locations at very early time points following irradiation? Since proliferation markers have not been measured, the conclusion that this population is locally self-renewing is not rigorously supported by data. A related but broader question is; how long do intravascular dextran-tagged perivascular macrophages persist in that location?

3) In light of reports that perivascular macrophages express Tie-2 and are highly pro-angiogenic and anti-inflammatory (see the work by de Palma et col.), and that Tie-2+ macrophages are known to be related to patrolling Cx3CR1 monocytes, it would be useful to have some further phenotypic characterization of these STREAM cells for such markers.

4) Isolating macrophages from skin requires harsh digestion. It would be useful to have evidence about how this affects cell surface phenotype, viability and other properties.

5) The data in Figure 7 should be accompanied by formal demonstration of co-localization, and its insets should be at a higher resolution.

6) The authors follow the schema of classically (IFN-γ) or alternatively (IL-4) activated macrophages. However, macrophage polarization is possible without these cytokines (Mills et col., 2000). Also, there is growing opinion that there is a complex spectrum of macrophage functionality, rather than the simplified M1/M2 duality (Ginhoux et al. Nat. Immunol. (2015); Immunity (2016)). Further, the manuscript argues for a simple binary distinction between 'pro-inflammatory' and 'anti-inflammatory', which is based on a handful of genes and not quite borne out by the data. Thus, while discussing Figure 6, the authors switch their labeling in the figures from donor/host to pro- and anti-inflammatory. It would also be important to test responses to ex vivo stimulation with IL-4 and look at transcriptional induction to complement the data on responses to LPS. It would be useful if the manuscript could be revised to reflect the complex functional landscape of macrophages.

7) The manuscript does not clearly specify the use of DTR-transgenic mice, yet DT beads are used to deplete STREAM cells. The issue should be clarified.

8) The authors interpret the data as supportive for 'cell-autonomous' programming; how is a lineage-specific cellular autonomy of programming being distinguished from micro-environmental distinctions provided by, say, exposure to intravascular-extravascular versus extravascular-alone microenvironments? In fact, what is the functional relevance of the microvascular protrusions of the STREAM cells? Is the lack of LPS-responsiveness in these 'STREAMs' related to 'training', mediated perhaps by constant exposure to intravascular content?

9) New data should not be introduced in the Discussion.

10) Do the authors have any evidence based on staining of human skin samples to show the expression of this subset there as well?

---

## [Author Response]

*Essential revisions:*

*1) It would be very useful to have data and clarity about the developmental origin of these cells, and the relationship of this macrophage group with other ontogenetically distinct subsets of dermal macrophages previously reported. Dual origins for skin macrophages have been previously reported as a CCR2-independent prenatally derived subset and a CCR2-dependent postnatal bone marrow-derived subset (Jakubzick et al. and Tamoutounour et al. 2013). Is the origin of these STREAM macrophages yolk-sac and/or fetal liver? Is their development c-Myb-dependent?*

We agree that understanding the developmental origin of STREAMs is fundamental to establish ontogenetic relationships with other dermal macrophage subsets. However, elucidating the ontogeny of these cells and their possible dependence on c-Myb are not trivial questions. A detailed ontogenetic analysis would entail the generation of reagents (such as acquisition and breeding of mouse strains necessary for the study) and extensive additional experimentation that cannot be achieved in a reasonable time frame. We have discussed a plausible developmental origin for STREAMs in the Discussion (fifth paragraph), but we truly believe that the proposed study deserves a complete new piece of work and it is out of the scope of this paper.

Alternatively, we have worked thoroughly to understand the origin and maintenance of STREAMs in adulthood, which is also a related and very relevant question that complements the rest of the findings presented in our paper (imaging, phenotypic, transcriptional, and functional analyses) performed in adult skin. To tackle this question, we have carried out a fate mapping analysis of adult stem cells finding that STREAMs are renewed from a local progenitor not related to BM hematopoiesis (new Figure 5 and Figure 5—figure supplement 1 and Figure 5—figure supplement 2). We very much hope that the reviewers will be satisfied with this study that sheds new light on the origin of STREAMs at a post-natal stage.

Are these cells the same as or different from the perivascular macrophages that mediate neutrophil chemo-attraction (Abtin et al.)?

We have already addressed this question in the original version of the manuscript using the DPE-GFP strain, in which a subset of perivascular macrophages express GFP and mediate neutrophil chemo-attraction (reported by Abtin et al., Nat Immunol 2014). Combining microscopy and FACs analyses in the DPE-GFP animals treated with HMw TRITC-dextran, we reached the conclusion that STREAMs and GFP_+_ perivascular macrophages are mostly different macrophage subsets. In addition, congenic wt BM chimerism into DPE-GFP recipients demonstrated that, conversely to STREAMs, the majority of DPE-GFP macrophages are radio-sensitive (Figure 4).

*2) The authors refer to this macrophage subset as 'radio-resistant' and suggest that these cells are maintained locally. These are two separate properties though perhaps related, and require elucidation. Thus, does 'radio-resistance' refer to the differentiated perivascular cells themselves, or to some progenitor population?*

We now provide data demonstrating that STREAMs are renewed and maintained locally from a radio-resistant progenitor, using a lineage tracing analysis in adult tissue (new Figure 5 and Figure 5—figure supplement 2). In particular, we have generated chimeric animals using *Bmi1*-IRES-Cre-ERT2 Rosa26YFP CD45.2 reporter animals as recipients and congenic CD45.1 C57BL/6 wt as donor mice.

The *Bmi1* gene is involved in the maintenance of adult stemness and is expressed by stem cells and multipotent progenitors, but is barely detectable in differentiated macrophages (Figure 5—figure supplement 2). After tamoxifen treatment, we have been able to detect YFP^+^ dextran^+^ macrophages in the skin of these chimeric mice, excluding the contribution of BM-derived HSC to STREAM renewal and demonstrating the existence of a skin radio-resistant pluripotent progenitor Bmi1^+^ that gives rise to STREAMs (Figure 5).

*Can comparable numbers of STREAM cells be detected in perivascular locations at very early time points following irradiation?*

To address this question, we have included new data in Figure 3—figure supplement 1 showing the presence of comparable numbers of dextran^+^ macrophages in a mouse ear 7d after lethal γ-irradiation by intravital imaging.

*Since proliferation markers have not been measured, the conclusion that this population is locally self-renewing is not rigorously supported by data.*

We now provide data showing that STREAMs are unable to proliferate in steady-state using Ki-67 as proliferation marker (Figure 5). However, STREAMs experience a slow turnover observed in vivowith BrdU labeling (Figure 5). This turnover cannot be explained by self-renewal based in Ki-67 data but by the existence of a progenitor from which they are renewed (lineage tracing analysis already commented, Figure 5).

A related but broader question is; how long do intravascular dextran-tagged perivascular macrophages persist in that location?

Using intravital microscopy, we have monitored dextran^+^ perivascular STREAMs up to 7d after dextran administration. Interestingly, long-term studies have reported the detection of dextran_+_ macrophages in the skin 2 months after intradermal injection of 155 kDa dextran (Intravital Immunofluorescence for Visualizing the Microcirculatory and Immune Microenvironments in the Mouse Ear Dermis. Kilarski W.W., Güc E., Teo J.C., Oliver S.R., Lund A.W., Swartz M.A. PLOS One2013;8(2): e57135).

3) In light of reports that perivascular macrophages express Tie-2 and are highly pro-angiogenic and anti-inflammatory (see the work by de Palma et col.), and that Tie-2+ macrophages are known to be related to patrolling Cx3CR1 monocytes, it would be useful to have some further phenotypic characterization of these STREAM cells for such markers.

Following the reviewers’ suggestion, we have analysed the expression of Tie-2 in skin BM-derived macrophages and STREAMs not finding differences (data included in new Figure 6—figure supplement 1). We have also performed intravital experiments using CX3CR1*^gpf/+^*mice that show the lack of CX3CR1expression in dextran^+^ STREAMs (data included in Figure 3—figure supplement 1).

4) Isolating macrophages from skin requires harsh digestion. It would be useful to have evidence about how this affects cell surface phenotype, viability and other properties.

We use a standard cell isolation protocol to homogenize mouse skin based on liberase TM digestion, which preserves all the cell surface markers used in this study (Preparation of Single-cell Suspensions for Cytofluorimetric Analysis from Different Mouse Skin Regions. Broggi A., Cigni C., Zanoni I., Granucci F. J Vis Exp. 2016; 110). As requested, we have included an example of the viability obtained using this protocol (Figure 6—figure supplement 1, DAPI staining – viability ≈ 65%).

5) The data in Figure 7 should be accompanied by formal demonstration of co-localization, and its insets should be at a higher resolution.

As requested by the reviewers, we provide a 3D colocalization mask of the green and red channels in new Figure 8 (former Figure 7) and enlarged insets. The complementary FACs analysis shown in Figure 8—figure supplement 1 confirms the existence of dextran^+^ macrophages containing OVA-FITC Fluospheres.

6) The authors follow the schema of classically (IFN-γ) or alternatively (IL-4) activated macrophages. However, macrophage polarization is possible without these cytokines (Mills et col., 2000). Also, there is growing opinion that there is a complex spectrum of macrophage functionality, rather than the simplified M1/M2 duality (Ginhoux et al. Nat. Immunol. (2015); Immunity (2016)). Further, the manuscript argues for a simple binary distinction between 'pro-inflammatory' and 'anti-inflammatory', which is based on a handful of genes and not quite borne out by the data. Thus, while discussing Figure 6, the authors switch their labeling in the figures from donor/host to pro- and anti-inflammatory. It would also be important to test responses to ex vivo stimulation with IL-4 and look at transcriptional induction to complement the data on responses to LPS. It would be useful if the manuscript could be revised to reflect the complex functional landscape of macrophages.

We totally agree with this reviewer on the existence of a very complex spectrum of macrophage polarization and functionality than cannot be explained by the simplified M1/M2 dichotomy. As our study has been made with macrophage subsets isolated under steady-state conditions, we have avoided the use of such M1/M2 nomenclature. The distinction of skin-resident macrophage subsets harboring pro-inflammatory or anti-inflammatory potential in steady-state was attained by combining not only the RNASeq analysis (metabolic programming), but the immune transcriptional profile and the expression of standard phenotypic markers. In addition, the ex vivostimulation with LPS and IL-4 was performed to evaluate the plasticity of STREAMs. In fact, following the interesting comment raised by the reviewer, we now show that STREAMs produce IL-10 after IL-4 stimulation (new Figure 7), demonstrating that they are responsive to an anti-inflammatory stimulus but refractory to a pro- inflammatory challenge (LPS). These results further emphasize the lack of plasticity of STREAMs and their anti-inflammatory commitment already in steady-state.

Regarding the labeling, we first termed the macrophage subsets as donor/host during the transcriptional profiling analyses and, once found their different profiles, we switched to the more intuitive pro-/anti-inflammatory terminology. Thus, we have found a binary distinction between radio-resistant anti-inflammatory and radio-sensitive pro-inflammatory macrophages, but we cannot rule out the existence of more heterogeneity and complexity within each subset. All these issues have been clarified throughout the text to better reflect the complex functional landscape of macrophages.

7) The manuscript does not clearly specify the use of DTR-transgenic mice, yet DT beads are used to deplete STREAM cells. The issue should be clarified.

We have not used CD11b–DTR animals or other available DTR strains to deplete macrophages after DT administration, because this strategy does not allow for a selective depletion of STREAMs (all skin-resident macrophages would be targeted). Alternatively, we have taken advantage of the specific ability of STREAMs to uptake blood-borne OVA nanoparticles. These OVA-NP were co-coated with DT (DT-OVA- NP) and administered i.v. In that manner, STREAMs were the only skin macrophages that could get access and capture circulating DT-OVA-NP, becoming poisoned after ingestion and being selectively depleted (explained in the subsection “Selective depletion of STREAMs impairs skin repair”, second paragraph).

*8) The authors interpret the data as supportive for 'cell-autonomous' programming; how is a lineage-specific cellular autonomy of programming being distinguished from micro-environmental distinctions provided by, say, exposure to intravascular-extravascular versus extravascular-alone microenvironments?*

The LPS and IL-4 treatments performed ex vivoin sorted macrophage populations could support the existence of a cell-autonomous programming that allows a differential polarization of anti-inflammatory STREAMs vs. pro-inflammatory skin macrophages in the absence of differential environmental cues. However, we agree with the reviewer in pointing out that the access to the intravascular milieu could provide STREAMs with unique signals that could cooperate on their programming. This fact has been now reflected in the text (Introduction, last paragraph and Discussion, eighth paragraph).

In fact, what is the functional relevance of the microvascular protrusions of the STREAM cells?

The ability of STREAMs to protrude across endothelial junctions and reach the vascular lumen to scavenge blood-borne fluorescent dextran allowed us to discover and characterize this subset, distinguishing them from other skin-resident macrophages. In our present study, we have focused on the relevant anti- inflammatory properties and functions of STREAMs. The analysis of the function of their intraluminal protrusions is a very interesting question and our preliminary data suggest a role for STREAMs in capturing and processing circulatory antigens to cross-present them to tissue-resident memory CD8^+^ T cells. However, the elucidation of this function deserves much further characterization and should be considered as a complete new piece of work.

Is the lack of LPS-responsiveness in these 'STREAMs' related to 'training', mediated perhaps by constant exposure to intravascular content?

The amount of bioactive LPS in the blood of healthy untreated animals should be negligible and insufficient for “training”. Moreover, assuming this argument is right, circulating neutrophils and monocytes should become also “trained” and be refractory to a potent LPS stimulation, but it is well known that these cells are very responsive instead (Lipopolysaccharide-induced expression of cell surface receptors and cell activation of neutrophils and monocytes in whole human blood. Gomes N.E., Brunialti M.K., Mendes M.E., Freudenberg M., Galanos C., Salomão R. Braz J Med Biol *Res.* 2010; 43(9):853-8).

9) New data should not be introduced in the Discussion.

The unpublished observations commented in the Discussion have been deleted from the text and the data not shown in are now included as new Figure 5.

*10) Do the authors have any evidence based on staining of human skin samples to show the expression of this subset there as well?*

STREAMs can be distinguished from others skin macrophages due to their radio- resistant nature and their capacity to uptake blood-borne fluorescent macromolecules. Neither of these properties can be directly analyzed in a human model. Moreover, the absence of a subset-defining marker precludes a straightforward FACs analysis. For this reason, we have performed and present for the reviewers an immunofluorescence analysis of human skin sections stained for CD31 (red, vasculature) and CD163 (green, macrophage marker widely used in human samples – CD163: A Specific Marker of Macrophages in Paraffin-Embedded Tissue Samples. Lau S.K., Chu P.G., and Weiss L.M. Am J Clin Pathol2004;122:794-801). The perivascular CD163^+^ macrophages in close contact to endothelial cells, some of them even protruding with cellular projections into the endothelial layer, have been highlighted in white boxes. This is a correlation with our observations made in the mouse model, but due to the inability to perform further functional analyses in human skin we cannot unequivocally conclude that these cells are the human homologues of STREAMs.

Author response image 1.Skin biopsies were collected, immediately embedded in OCT and frozen.Then, the tissue was cut in 30 μm-thick criosections. After fixation and permeabilization, skin sections were blocked with human-γ globulin and 5% FBS. Then, sections were incubated with mouse anti-human CD31 for 1 hr followed by incubation with Alexa-Fluor 555 donkey anti-mouse. Next, samples were blocked with mouse serum 1:50 O/N 4°C, subsequently mouse-anti human biotinylated anti-CD163 at 5 μg/ml was added for 1 hr and, finally, samples were incubated with Streptavidin Alexa-Fluor 488. DAPI staining was performed just before mounting the sample for microscopy observation using a standard confocal microscope.**DOI:**
http://dx.doi.org/10.7554/eLife.15251.035